# Remarks on Berry connection in QFT, anomalies, and applications

**Mykola Dedushenko**

Simons Center for Geometry and Physics, Stony Brook University,
Stony Brook, NY 11794-3636, USA

## Abstract

Berry connection has been recently generalized to higher-dimensional QFT, where it can be thought of as a topological term in the effective action for background couplings. Via the inflow, this term corresponds to the boundary anomaly in the space of couplings, another notion recently introduced in the literature. In this note we address the question of whether the old-fashioned Berry connection (for time-dependent couplings) still makes sense in a QFT on $\Sigma^{(d)} \times \mathbb{R}$, where $\Sigma^{(d)}$ is a $d$-dimensional compact space and $\mathbb{R}$ is time. Compactness of $\Sigma^{(d)}$ relieves us of the IR divergences, so we only have to address the UV issues. We describe a number of cases when the Berry connection is well defined (which includes the $tt^*$ equations), and when it is not. We also mention a relation to the boundary anomalies and boundary states on the Euclidean $\Sigma^{(d)} \times \mathbb{R}_{\geq 0}$. We then work out the examples of a free 3D Dirac fermion and a 3D $\mathcal{N} = 2$ chiral multiplet. Finally, we consider 3D theories on $\mathbb{T}^2 \times \mathbb{R}$, where the space $\mathbb{T}^2$ is a two-torus, and apply our machinery to clarify some aspects of the relation between 3D SUSY vacua and elliptic cohomology. We also comment on the generalization to higher genus.

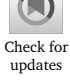

# 1 Introduction

Berry connection provides one of the simplest links between quantum mechanics and topology, as formulated by Berry [1] and Simon [2]. The construction involves a Hilbert space $\mathcal{H}$ and a space of parameters $M$, such that the Hamiltonian $H$ depends on the point of $M$, while the Hilbert space does not. One picks an eigenstate (usually the vacuum) or, more generally, a finite-dimensional eigenspace of $H$, separated by gaps from the rest of spectrum for all values of the parameters. This determines a finite-dimensional subbundle $\mathcal{V}$ inside the trivial bundle $M \times \mathcal{H}$, with the embedding denoted by I : $\mathcal{V} \to M \times \mathcal{H}$. The spectral projector onto the chosen eigenspace gives another map[1] $\Pi : M \times \mathcal{H} \to \mathcal{V}$. The bundle $\mathcal{V}$ is naturally equipped with the projector connection, known in this context as the Berry connection, defined by $\nabla = \Pi \cdot \mathrm{d} \cdot \mathrm{I}$, where d is a trivial connection on $M \times \mathcal{H}$. This construction was originally formulated in a completely finite-dimensional context of a spin in the magnetic field, but it works more generally, whenever the two conditions are met. First, $\mathcal{H}$ must be trivially fibered over $M$, with the trivialization $M \times \mathcal{H}$ fixed [3] for the trivial connection d to make sense at all, and second, the connection d on $M \times \mathcal{H}$ must be well-defined (not take us "outside" the Hilbert space). Both of these conditions are met in the ordinary quantum-mechanical systems with finite number of degrees of freedom, even though the Hilbert space may be infinite-dimensional in general.

Physically, the Berry connection captures evolution of a gapped state under the adiabatic change of parameters [4]. By this one means that the parameters move from point $a \in M$ to $b \in M$ so slowly that only some sort of "leading" behavior matters. This can be made more precise in the language of effective field theory. Let $\phi : \mathbb{R} \to M$ describe *background fields* in 0+1D, i.e., the change of parameters with time. Integrating out all the dynamical degrees of freedom produces an effective action for $\phi(t)$ that describes the vacuum response:

$$\Gamma[\phi] = \int \mathrm{d}t \left[ a^{(0)}(\phi) + a_i^{(1)}(\phi)\dot{\phi}^i + a_{ij}^{(2)}\dot{\phi}^i\dot{\phi}^j + \dots \right]. \tag{1}$$

Here we assume the answer to be local, and write it in the form of derivative expansion. If the change of parameters happens over the time $T$, the first term in (1) scales as $T$, the second one is $T$-independent, and the remaining terms scale as negative powers of $T$. In the limit of large $T$, only the first two terms survive. The first, extensive, term describes the energy of zero point fluctuations. It is generically present, and its contribution is usually easy to account for and subtract. Either by doing so or by focusing on supersymmetric theories (in which the vacuum

---

[1]Note that $\Pi \cdot \mathrm{I} = \mathrm{Id} \in \mathrm{End}(\mathcal{V})$, and all the nontrivial data is contained in $\mathrm{I} \cdot \Pi = P \in \mathrm{End}(\mathcal{H})$, which is a projector-valued function on $M$.

energy vanishes, unless SUSY is broken), we isolate the second term. It can be written as

$$\Gamma_{\text{top}} = \int_{\gamma} a_i^{(1)}(\phi)\mathrm{d}\phi^i \,, \tag{2}$$

where $\gamma \subset M$ is a path connecting $a$ and $b$ in the parameter space. This term is $T$-independent, or equivalently, independent of the worldline metric, which is why $\Gamma_{\text{top}}$ is called the topological term. The $T \to \infty$ limit is precisely the adiabatic limit, and as we see it singles out the first two terms in (1). The subleading topological term contains a one-form (in fact, a gauge field) $a^{(1)}$ on $M$, which is precisely the Berry connection. When $\gamma$ is a closed loop, $\Gamma_{\text{top}}$ is known as the Berry phase.

It is natural to wonder how this generalizes to systems with infinitely many degrees of freedom, such as quantum fields theories (QFT) or many-body systems, and the recent papers [5–9] were asking just that. The applicability conditions of the usual Berry connection construction could break in this case. It may happen in QFT that the Hilbert bundle is not naturally trivialized, basically due to the UV issues. Even if it is trivialized, the trivial connection d on $M \times \mathcal{H}$ may suffer from the IR divergences in a noncompact space, both in QFT and many-body systems. In [9], the first issue is bypassed via the algebraic approach that does not rely on the Hilbert space, and the second one is resolved [6, 9] by constructing an IR-finite $(D + 1)$-form analog of the Berry curvature in $D$ spacetime dimensions. The authors of those works take the case of many-body systems more seriously, even though their constructions seem to apply to QFT as well.

The effective field theory considerations on $X = \mathbb{R}^D$ in fact do suggest to look for a $D$-form Berry connection. By analogy with 1D, it captures topological terms in the effective action that describe the vacuum response to adiabatic *spacetime* (rather than just time) variations of the parameters. In $D > 1$ spacetime dimensions, a possible topological term may be written as an integral of a pullback by $\phi : X \to M$ of a $D$-form $\omega$ on $M$:

$$\Gamma_{\text{top}} = \int_X \phi^* \omega \,, \tag{3}$$

which more generally is a $D$-form gauge field on $M$. This is a higher-form generalization of the Berry phase, according to [6]. One can also activate a background gauge field $A$ on spacetime to write more general topological terms, a simple one being the Thouless pump [7, 10–12],

$$\int_X A \wedge \phi^* \tau \,, \tag{4}$$

where $\tau$ is a closed $(D - 1)$-form on $M$ with quantized periods. Whenever $A$ over $X$ can be extended to $W_{D+1}$, such that $X = \partial W_{D+1}$, the latter term is written as

$$\int_{W_{D+1}} F \wedge \phi^* \tau \,, \tag{5}$$

where $F = \mathrm{d}A$. More general topological terms are built from the closed forms and characteristic classes on $W_{D+1}$. Since the characteristic classes of a $G$-bundle are universally pulled back from $H^*(BG)$, such topological terms are classified by

$$H^{D+1}(BG \times M) = \bigoplus_{p+q=D+1} H^p(BG) \otimes H^q(M), \tag{6}$$

where $H^{D+1}(BG)$ labels the pure Chern-Simons terms, the Thouless pump term lives in $H^2(BG) \otimes H^{D-1}(M)$, and so on. Slightly more generally, the space of parameters $M$ could

be acted on by $G$, in which case the topological terms are classified by the equivariant cohomology group[2]

$$H_G^{D+1}(M) \equiv H^{D+1}(EG \times_G M). \tag{7}$$

In fact, the construction of an invariant in $H_G^{D+1}(M)$ for lattice systems was given in [9]. Such an invariant of course captures the Berry *curvature*, thus ignoring flat connections and possible torsion effects.

A few more generalizations are possible, such as incorporating the background metric (which, in generally covariant theories, amounts to replacing $BG$ by $BG \times B\mathrm{Spin}_D$ in the above) and higher-form symmetries. Also, an extension to $W_{D+1}$ is not always available, so general topological terms are constructed via differential cohomology, as explained in [13, 14]. Most of these questions are well-understood by now, since the classification of possible topological terms is the same as classification of the deformation classes of invertible topological field theories (TFT) [15], or classification of the generalized anomaly inflow terms in the sense of [13, 14]. They are labeled by the Anderson dual of the appropriate bordism theory [15], see also [16–21]. Its torsion part classifies global anomalies in $(D-1)$ dimensions, while the free part corresponds to topological terms of the kind we discussed so far. In this paper we are only interested in the free part, since it is directly related to the Berry curvature. For example, if we include: (1) background gauge fields for the global 0-form symmetry $G$, (2) background metric, and (3) spatially-modulated $G$-neutral scalar parameters (couplings) valued in $M$, then the corresponding topological term is labeled by

$$H^{D+1}\left(BG \times B\mathrm{Spin}_D \times M\right). \tag{8}$$

It has the meaning of higher Berry curvature in the $D$-dimensional theory, where the said parameters are adiabatically varied in spacetime. Alternatively, it is understood as the anomaly polynomial for a $(D-1)$-dimensional theory, in which "anomalies in the space of couplings" are taken into account. Such $(D-1)$-dimensional theories can be realized, for example, at the boundaries or interfaces of the former $D$-dimensional theory, receiving their anomaly through the inflow mechanism.

The notion of generalized $D$-form Berry connection in a $D$-dimensional QFT does not, however, close the subject of old-fashioned Berry phase in QFT [5, 22]. It would be problematic to simply claim that the corresponding one-form Berry connection is ill-defined whenever $D > 1$. Indeed, it has spectacular applications in SUSY field theories via the tt* equations. They describe the geometry of vacuum bundle, originally in 2d $\mathcal{N} = (2, 2)$ theories [23], and later for their 3d and 4d uplifts [24–27]. Recently, similar constructions were used in [28, 29] in the context of relating 3d $\mathcal{N} = 2$ theories on an elliptic curve to the elliptic cohomology.[3] The goal of this note is to clarify the applicability of the "old" Berry phase in QFT, and to solidify certain aspects of it that appeared recently in [28, 29].

It is beyond doubt that the two-form Berry curvature on $X = \mathbb{R}^D$ is IR divergent for $D > 1$ [6]. It would be captured by the term $a_i^{(1)}(\phi)\dot{\phi}^i$ in the effective action, and if the couplings only vary in time, such a term is invariant under spatial translations leading to the IR divergence. This issue is straightforward to overcome by only varying the couplings $\phi$ in a bounded region of space, or alternatively by placing QFT on

$$\Sigma^{(D-1)} \times \mathbb{R}, \tag{9}$$

with some compact spatial slice $\Sigma^{(D-1)}$. At large distances such a theory looks one-dimensional, and we again study its one-dimensional effective action (1). The Berry phase term $a_i^{(1)}(\phi)\dot{\phi}^i$

---

[2]If the global symmetry does not act on parameters, then of course $EG \times_G M = BG \times M$.

[3]In fact, [29] apply the machinery of tt* equations of [25], while [28] rely on the Berry connection for flat gauge fields only, as will be elaborated later in this paper.

then becomes perfectly IR-finite.[4] This is enough to make it well-defined in the many-body contexts, for example in a quantum Hall system on a torus (see, e.g., lecture notes [30]). In QFT, however, we still have to deal with the UV issues. In a renormalizable theory, we generally expect to get a UV-finite effective action, but the ambiguities present in general backgrounds might render the answer regularization scheme dependent, i.e., sensitive to details of the UV physics.

The question one should be asking, therefore, is whether the 1D effective action for time-dependent parameters $\phi : \mathbb{R} \to M$ of a D-dimensional theory placed on $\Sigma^{(D-1)} \times \mathbb{R}$ contains unambiguous terms of the form $a_i^{(1)}(\phi)\dot{\phi}^i$. More generally, we can put a theory on

$$\Sigma^{(D-p)} \times \mathbb{R}^p \,, \tag{10}$$

and study the $p$-form Berry connection as a topological term in the effective action on $\mathbb{R}^p$. In this case, all lower-form connections will be IR divergent. We do not consider such a more general setup in this paper and only focus on the case $p = 1$.

It is thus clear that the question of whether the Berry connection is well-defined boils down to a simple and well-known problem in effective field theory: The classification of finite counterterms in a given background. Namely, one should check if a UV theory on $\Sigma^{(D-1)} \times \mathbb{R}$ admits any local $D$-dimensional counterterms (preserving the symmetries), which in a time-dependent background $\phi : \mathbb{R} \to M$ contribute nontrivially to the term $a_i^{(1)}(\phi)\dot{\phi}^i$ in the 1D effective action. The absence of such finite UV counterterms serves as a proof that the Berry connection is unambiguous. On the contrary, if the finite counterterms do exist, the standard naturalness considerations imply that they are generated, sensitive to the UV completion, and render the answer (i.e., the effective action) ambiguous. Such a reasoning is used in the literature to extract the physical content of observables in non-trivial backgrounds (see [31, 32]). Note that we should be careful with the naturalness arguments, as one learns for example from [33], where a naively allowed interaction is found to be excluded by more exotic symmetries. Once the counterterms are excluded, however, this firmly establishes the uniqueness of the answer.

Note that the finite counterterms themselves can sometimes include the inflow actions, such as the Chern-Simons (CS) terms in 3D. The latter may be generated, e.g., by some heavy fermions living above the UV scale and decoupled from the IR physics. Such counterterms are relatively mild and can be eliminated by the proper choice of a regularization scheme. This is straightforwardly seen if the theory *admits boundary conditions*. Boundary conditions can be viewed as an interface between our theory and an empty space, and it is natural to require that the empty space has zero action, in particular no background inflow terms. This fixes the said ambiguity, since adding a background anomaly inflow term on the non-empty side would contribute to the boundary anomaly, which is not ambiguous. For example, such considerations allow to state that a Dirac fermion in 3D with real mass $m$ contributes the well-known CS level $\frac{1}{2}\mathrm{sign}(m)$ for its $U(1)$ symmetry [34–36], and not just $\frac{1}{2}\mathrm{sign}(m) + n$ with some $n \in \mathbb{Z}$.

We will also consider a closely related setup of Euclidean QFT on

$$\Sigma_{D-1} \times [0, +\infty) \,, \tag{11}$$

where the (Euclidean) time $\mathbb{R} \ni y$ is replaced by a half-line with some boundary condition $\mathcal{B}$ at $y = 0$. It is represented by a boundary state $|\mathcal{B}\rangle$, which encodes the topological data of

---

[4]When the IR divergence is treated in this way, the expected contribution to $a_i^{(1)}(\phi)$ is $\propto \mathrm{Vol}(\Sigma^{(D-1)})$. More generally, the volume does not have to appear linearly: $a_i^{(1)}(\phi)\dot{\phi}^i$ may be the 1D reduction of terms in the effective action on $\Sigma^{(D-1)} \times \mathbb{R}$ with nontrivial dependence on the metric of $\Sigma^{(D-1)}$. Also $a_i^{(1)}(\phi)$ may be volume-independent, as in Section 3, where it descends from the topological (Chern-Simons) terms in 3D.

boundary 't Hooft anomalies supported by $\mathcal{B}$, including the anomalies in the space of couplings. The way it works is similar to how the topological data of the vacuum bundle is carried by the Berry connection, which is of course natural given that the Berry connection terms in the bulk effective action induce inflow for the boundary anomalies. These relations will be demonstrated in the examples later in this paper.

We start the discussion in Section 2 by analyzing the background counterterms in a number of examples. The background fields in this analysis include scalar couplings, gauge fields, and the metric. Considering examples of progressing complexity, we identify when the corresponding Berry connection is well-defined. We also separately discuss an important case of tt* geometry, for which we also check that the Berry connection is well-defined.

In Section 3 we illustrate some of these ideas with an example of free massive Dirac fermion in 3D, as well as its $\mathcal{N} = 2$ generalization — a chiral multiplet. The 3D spacetime is $\mathbb{T}^2 \times \mathbb{R}$, where $\mathbb{T}^2$ is a flat two-torus, and the parameters include the flat $U(1)$ connection on $\mathbb{T}^2$ and the real mass. We show that the classic definition of Berry connection can be made sense of in this case, and the results agree with our expectations. We also consider the geometry $\mathbb{T}^2 \times \mathbb{R}_{\geq 0}$ with an elliptic boundary condition $\mathcal{B}$ imposed at Euclidean time 0, and demonstrate how the boundary state $|\mathcal{B}\rangle$ captures the boundary anomaly.

In Section 4 we consider a general (trivially) gapped 3D QFT on $\mathbb{T}^2 \times \mathbb{R}$ or $\Sigma_{g \geq 2} \times \mathbb{R}$, where $\Sigma_g$ is a genus $g$ Riemann surface. We explain that the effective CS terms capture the Berry connection for background gauge fields on $\Sigma_g$, and in particular we focus on the flat background gauge fields. Their Berry connection determines the holomorphic structure on the vacuum bundle, and we compute its holomorphic sections. We then note that supersymmetric partition functions on $\mathbb{T}^2$ or $\mathbb{T}^2 \times I$, where $I$ is an interval, provide such holomorphic sections. This goes back to, e.g., [37], and we explicitly use the result of [38].

Finally, in Section 5 we apply all of these to clarify the relation between 3d $\mathcal{N} = 2$ gauge theories on $\mathbb{T}^2 \times \mathbb{R}$ and elliptic cohomology. The one sentence summary of this relation can be formulated as follows: While the holomorphic bundle of vacua makes sense for any symmetric gapped QFT on $\mathbb{T}^2 \times \mathbb{R}$, in the SUSY case the supersymmetric partition function provides its holomorphic section, interpreted as an elliptic cohomology class on the moduli space of vacua (which also only exists in the SUSY context). An important ingredient in our analysis is a careful computation of the effective CS terms on the Higgs branch, both with and without the real mass deformation. We then comment on the higher genus generalization.

## 2 Finite counterterms

In the rest of this paper we consider a $D$-dimensional QFT on a compact space $\Sigma^{(D-1)}$,

$$\Sigma^{(D-1)} \times \mathbb{R}, \tag{12}$$

and study the term $a_i^{(1)}(\phi)\dot{\phi}^i$ in the 1D effective action, where $\phi^i$ are parameters promoted to background fields. For many-body systems, the long-distance effective QFT description is unambiguous, and identifying the UV-IR map is one of the central tasks. When we ask the same questions in a renormalizable QFT, we are looking for terms in the effective action that are intrinsic to the continuum description and independent of the UV completion. Such universal terms are defined modulo local background counterterms allowed in the UV. In our case, we must classify *finite* counterterms that contribute to $a_i^{(1)}(\phi)$ in the effective action. If such counterterms are present, they make the Berry connection ill-defined. The "positive" result is when they all vanish, implying that the Berry phase is well-defined.

The term $a_i^{(1)}(\phi)\dot{\phi}^i$ itself is a valid finite counterterm in 1D. However, it does no harm because there are no UV issues in quantum mechanics, so the original definition of Berry

connection [1, 2] is finite and unambiguous. In a $D$-dimensional QFT on $\Sigma^{(D-1)} \times \mathbb{R}$ with $D \geq 2$, we should look for the $D$-dimensional counterterms that could reduce to $a_i^{(1)}(\phi)\dot{\phi}^i$. Let us try to construct them from the background fields, which for us include:

1. Scalars $\phi$, i.e., couplings promoted to background fields,

2. the metric,

3. gauge fields for flavor symmetries.

(Sometimes $\phi$ will denote all these parameters collectively.) Such counterterms, in the modern language, correspond to invertible field theories. Their classification is known, and in particular classification modulo continuous deformations (i.e., deformation classes) corresponds to the classification of anomaly inflow actions or SPT phases [15–17] . We will consider counterterms in the trivial deformation class (i.e., deformable to zero) as "unprotected", while the counterterms corresponding to non-trivial deformation classes as "protected" by anomalies. This is related to something that was already mentioned in the introduction: Both types of counterterms can appear due to the unknown UV physics, but if the theory admits boundary conditions, there is a way to partially fix the ambiguity via anomalies. Indeed, boundary conditions carry some unambiguous boundary anomalies. The counterterms could contribute via the inflow, but since they are ambiguous, they should not, and indeed they do not contribute as they live on both sides of the boundary (their coefficients cannot jump in space). Thus we can simply demand that the "empty" side of the boundary has zero inflow action. This uniquely fixes the deformation class of the invertible field theory on the "filled" side of the boundary. In the simplest example, such class is labeled by the effective CS level in a given gapped vacuum of a 3D QFT. The remaining freedom only corresponds to the unprotected finite couterterms, i.e., those in the trivial deformation class.

## 2.1 Classification in flat space

Throughout this subsection, we consider the space to be a flat torus $\Sigma^{(D-1)} = \mathbb{T}^{D-1}$.

*Scalar couplings in flat space*

We start with the case of no background gauge fields. In the context of "old" Berry connection, we make $\phi$ time-modulated, while keeping them constant along the spatial slice $\Sigma^{(D-1)}$. Therefore, the spatial derivatives of $\phi$ vanish and we only have $\phi$ and its time derivatives in our disposal. Furthermore, we should have precisely one time-derivative $\dot{\phi}$ in the expression, since the terms with more time derivatives vanish in the adiabatic limit, and those without time derivatives are of no relevance for us. Thus $a_i(\phi)\dot{\phi}^i$ is the only possibility, but since we are in $D > 1$ dimensions now, it should be promoted to a local Lorentz-invariant expression. In the absence of curvature or background gauge fields, this is clearly impossible.

   This argument shows that the old Berry connection for time-dependent scalar couplings is well-defined on $\mathbb{T}^{D-1} \times \mathbb{R}$, simply because there are no local counterterms that would render the term $a_i^{(1)}(\phi)\dot{\phi}^i$ UV-sensitive. Notice that it is important to keep background gauge fields at zero, for even a flat background connection can invalidate this argument, as we will see.

*Scalar couplings in flat space and fixed gauge background*

Let us now include time-independent background gauge fields on the $\Sigma^{(D-1)}$ for zero-form symmetries.[5] This enlarges the class of possible counterterms. For example, an abelian field $A$

---

[5]It is straightforward to generalize to the higher-form symmetries, but we skip it for brevity.

allows to write the Thouless charge pump term in two dimensions [7, 10–12],

$$\int_{S^1 \times \mathbb{R}} A \wedge \phi^* \tau, \quad [\tau] \in H^1(M, \mathbb{Z}). \tag{13}$$

Such a term is an examples of the equivariant higher Berry connection [9], here corresponding to the integral cohomology class $c_1 \times [\tau] \in H^2(BU(1), \mathbb{Z}) \times H^1(M, \mathbb{Z})$. It could be UV sensitive in general, but in a theory admitting boundary conditions, this topological term would give a non-zero inflow contribution to the boundary mixed anomaly between the $U(1)$ symmetry and the couplings $\phi$ [13]. Thus according to our general philosophy, the UV ambiguity is partially resolved by requiring that it agrees with the boundary anomaly. This only fixes the cohomology class $[\tau] \in H^1(M, \mathbb{Z})$, not the representative, hence leaving the residual freedom to shift $\tau$ by $df$. If we denote the background holonomy as $h = \int_{S^1} A$, the 1D limit of our topological term is

$$h \tau_i(\phi) \dot{\phi}^i, \tag{14}$$

where the cohomology class $[\tau]$ is fixed. The ambiguity of shifting $\tau$ by $df$ is harmless since it corresponds to the gauge transformation of the Berry connection. Indeed, different ways to regularize the Berry connection are expected to give gauge-equivalent answers here.

This has a straightforward generalization to $D = 2k + 2$ dimensions. Given a degree-$2p$ characteristic class $P_p[F]$ and the Chern-Simons $(2k - 2p + 1)$-form, we write the counterterm:

$$\int_{\Sigma^{(2k+1)} \times \mathbb{R}} P_p[F] \wedge \mathrm{CS}_{2(k-p)+1}[A] \wedge \phi^* \tau, \quad [\tau] \in H^1(M, \mathbb{Z}), \tag{15}$$

whose proper definition is via the differential cohomology, but an intuitive shortcut is to integrate $P_p[F] \wedge \mathrm{Tr}(F^{k-p+1}) \wedge \tau$ over the $2k + 3$ manifold bounded by $\Sigma^{(2k+1)} \times \mathbb{R}$. This term is again both an example of the higher Berry connection and, near the boundary, a source of mixed boundary anomaly involving the couplings $\phi$. Thus we can fix the ambiguity in the cohomology class $[\tau] \in H^1(M, \mathbb{Z})$ but not in its representative. Denoting

$$h_k = \int_{\Sigma^{(2k+1)}} P_p[F] \wedge \mathrm{CS}_{2(k-p)+1}[A], \tag{16}$$

the 1D reduction becomes $h_k \tau_i(\phi) \dot{\phi}^i$. Again, a shift $\tau \mapsto \tau + df$ corresponds to the gauge transformation of the Berry connection, which is a harmless ambiguity.

In $D = 2k + 1$ dimensions we have another class of counterterms:

$$\int_{\Sigma^{(2k)} \times \mathbb{R}} P_k[F] \wedge \phi^* \omega, \tag{17}$$

where $\omega$ is a one-form, or more generally a connection on $M$, and $P_k[F]$ is a characteristic class, such as $\mathrm{Tr}(F^k)$. This term is in general not quantized (at least when $\omega$ is a global one-form), it is not protected by the anomaly, so it leads to genuine ambiguity. There are three possibilities to get rid of it. One is to only consider topologically trivial $F$ on $\Sigma^{(2k)}$, such that all the characteristic classes $\int_{\Sigma^{(2k)}} P_k[F]$ vanish, implying that the counterterms vanish. Another possibility is for the couplings $\phi$ to have positive mass dimensions, so the above counterterm is not dimensionless and is suppressed by the inverse power of the UV cutoff. The third option is to have some symmetries that prohibit the offending counterterm. If these conditions are not met, the Berry phase for $\phi$ becomes ambiguous.

*Scalar couplings and time-dependent gauge fields in flat space*

Now consider the Berry connection both for the scalar couplings $\phi$ and for the background gauge fields (which are thus time-dependent). In the two-dimensional case, this activates the counterterm

$$\int_{S^1 \times \mathbb{R}} f(\phi)F\,, \tag{18}$$

which contributes $f(\phi)\dot{h}$ in the effective action, equivalent to the shift $\tau \mapsto \tau - \mathrm{d}f$ in the term $h\tau_i(\phi)\dot{\phi}^i$. This is no longer a gauge transformation, as $h$ is not a constant, so the Berry connection becomes truly ambiguous. The ambiguity can be resolved either when the counterterm is disallowed by symmetries, or if all the couplings $\phi$ have positive mass dimensions, such that the counterterm has positive mass dimension and is suppressed by the inverse power of the UV cutoff.

In $D = 2k + 2$ dimensions, similar arguments apply to the counterterm

$$\int_{\Sigma^{(2k+1)} \times \mathbb{R}} f(\phi)P_{k+1}[F]\,, \tag{19}$$

where $P_{k+1}[F]$ is some characteristic class, making the Berry connection generally ambiguous. In addition to the two previously mentioned possibilities, we can also resolve this ambiguity by restricting the class of allowed $F$ such that $P_{k+1}[F]$ vanishes. For example, for $k = 1$, $P_{k+1} = \mathrm{Tr}(F \wedge F)$ or $\mathrm{Tr}(F) \wedge \mathrm{Tr}(F)$, so it is enough to ask that the background gauge field is flat along space, $F\big|_{\Sigma^{(3)}} = 0$. This guarantees that the counterterm vanishes, and allows to unambiguously define the Berry connection for flat background gauge fields on $\Sigma^{(2k+1)}$.

In $D = 2k + 1$ dimensions when the gauge fields are time-dependent, we activate new background Chern-Simons counterterms, in general given by:

$$\int_{\Sigma^{(2k)} \times \mathbb{R}} P_p[F] \wedge \mathrm{CS}_{2(k-p)+1}[A]\,, \tag{20}$$

for some characteristic class $P_p[F]$. Such terms of course have quantized coefficients and provide non-trivial inflow contributions. Thus they are protected, and the ambiguity is fixed by asking that they agree with the boundary anomalies. Thus the Berry connection just for the background gauge fields can be made unambiguous in this case. Of course one still has to deal with the counterterms (17) if we vary couplings $\phi$ with time. In this case, as usual, the ambiguity is resolved either by symmetry requirements, or by making $F$ topologically trivial along $\Sigma^{(2k)}$, or by only considering $\phi$'s of positive mass dimension.

## 2.2 Metric along $\Sigma^{(D-1)}$ in diverse dimensions

Next include metric, i.e., let $\Sigma^{(D-1)}$ be a general Riemannian $(D-1)$-manifold.

If $D = 2$, then $\Sigma^{(D-1)} = S^1$ and its circumference $\beta$ is the only metric invariant. Keeping $\beta$ fixed adds nothing to the story and returns us to the previous case. If we allow $\beta$ to change with time, the 1D effective action for $\phi$ and $\beta$ can have two interesting terms:

$$a_i^{(1)}(\beta, \phi)\dot{\phi}^i + b(\beta, \phi)\dot{\beta}\,. \tag{21}$$

It is fairly clear that none of the 2D local diff-invariant expressions built out of $\phi$ and metric invariants can reduce to this (e.g., the Riemann tensor is proportional to $\ddot{\beta}$ here). Thus the Berry connection is still well-defined if we vary both $\phi$ and $\beta$ with time.

For $D = 3$, if we keep the metric of surface $\Sigma^{(2)}$ time-independent, there are no new harmful counterterms, and the flat-space analysis applies. In this case one can have divergent

counterterms like $\Lambda \int R\sqrt{g}\, \mathrm{d}^3 x$, where $\Lambda$ is the UV cutoff, but they do not contribute to the Berry connection of $\phi$ or $A$.

If we let the metric of $\Sigma^{(2)}$ vary with time, a new finite counterterm starts contributing:

$$\int_{\Sigma^{(2)} \times \mathbb{R}} \mathrm{CS}_{\mathrm{grav}}\,, \tag{22}$$

where $\mathrm{CS}_{\mathrm{grav}}$ is the gravitational Chern-Simons term that can be defined for example by

$$\int_{W_3 \times \mathbb{R}} \frac{1}{192\pi} \mathrm{Tr}\,(R \wedge R)\,, \tag{23}$$

where $W_3$ is a handle-body, such that $\partial W_3 = \Sigma^{(2)}$ (metric always extends smoothly to $W_3$). We cannot multiply $\mathrm{CS}_{\mathrm{grav}}$ by a function of $\phi$, so this term clearly does not affect the Berry connection $a_i^{(1)}(\phi)\mathrm{d}\phi^i$ for $\phi$. It does contribute to the Berry connection for the metric of $\Sigma^{(2)}$, but the ambiguity is fixed by demanding consistency with the boundary gravitational anomaly, since the gravitational CS contributes to it via the inflow.

The divergent counterterm $\Lambda \int_{\Sigma^{(2)} \times \mathbb{R}} f(\phi)R\sqrt{g}\, \mathrm{d}^3 x$ looks as if it could even make the Berry connection divergent, but this does not happen. The scalar curvature of $\Sigma^{(2)} \times \mathbb{R}$ with the time-dependent metric of $\Sigma^{(2)}$ is $R = R_2 - g^{\mu\nu}\partial_t^2 g_{\mu\nu} + \frac{1}{2}g^{\mu\nu}g^{\sigma\rho}\partial_t g_{\mu\sigma}\partial_t g_{\nu\rho}$, where $g_{\mu\nu}$ is the metric of $\Sigma^{(2)}$ and $R_2$ is its scalar curvature. This answer simply has no one-derivative terms. We thus conclude that in 3D, making the metric of $\Sigma^{(2)}$ curved and time-dependent does not invalidate the flat space analysis conducted previously, and furthermore, it even makes sense to study the Berry connection for the metric moduli of $\Sigma^{(2)}$.

For $D = 4$, we can write another finite counterterm:

$$\int_{\Sigma^{(3)} \times \mathbb{R}} \mathrm{CS}_{\mathrm{grav}} \wedge \phi^* \tau\,, \tag{24}$$

where $\tau$ is a closed one-form on $M$ with integer periods. Because $\phi$ only depends on time, $\phi^* \tau$ absorbs the $\mathbb{R}$ integration and the term factorizes:

$$\int_{\mathbb{R}} \phi^* \tau \int_{\Sigma^{(3)}} \mathrm{CS}_{\mathrm{grav}}\,, \tag{25}$$

making manifest that it contributes to $a_i^{(1)}(\phi)\dot{\phi}^i$ whenever $\frac{1}{2\pi}\int_{\Sigma^{(3)}} \mathrm{CS}_{\mathrm{grav}} \in \mathbb{R}/\mathbb{Z}$ is nontrivial. The term (24) is a secondary invariant that provides a nonzero inflow for the mixed gravity-coupling anomaly whenever a boundary is present. Because of that, we fix the discrete ambiguity labeled by $[\tau] \in H^1(M, \mathbb{Z})$ by demanding consistency with the boundary anomaly. It leaves the ambiguity $\tau \mapsto \tau + \mathrm{d}f$ that, as before, amounts to a gauge transformation of the Berry connection (if the metric of $\Sigma^{(3)}$ is static) and is harmless.

If we let the metric of $\Sigma^{(3)}$ change with time, such that its CS invariant becomes time-dependent, the counterterm (25) with $\tau = \mathrm{d}f$ no longer corresponds to the gauge transformation of the Berry connection. Rather, it represents a true ambiguity of the latter, unless we are unable to write such a counterterm for dimension reasons.[6] Furthermore, we earn a few new finite counterterms (with dimensionless $f(\phi)$): $f(\phi)\mathrm{Tr}\,(R \wedge R)$, $f(\phi)R^{\mu\nu\sigma\rho}R_{\mu\nu\sigma\rho}$, $f(\phi)W^2$, and $f(\phi)E_4$, where $W$ is the Weyl tensor and $E_4 = \frac{1}{2}\varepsilon^{abcd}\varepsilon^{pqrs}R_{abpq}R_{cdrs}$ is the Euler density. One can check that the $R^2$, $W^2$ and $E_4$ counterterms lack pieces with precisely

---

[6]Namely, we need a function $f(\phi)$ to be dimensionless and *not* constant, which is easy to achieve if $\phi$ itself is a dimensionless coupling, but may be impossible otherwise. Note: Using the UV cutoff to compensate the dimension is always possible but results in non-finite counterterms.

one time derivative, so they cannot contribute to the Berry term. The only counterterm that can potentially contribute is $f(\phi)\text{Tr}(R \wedge R)$. It is an invertible TFT in the zero deformation class that does not generate any inflow. Therefore, according to our logic, it signals about the true scheme-dependence and ambiguity in the Berry connection for the metric of $\Sigma^{(3)}$. This ambiguity is subject to the same caveat of the footnote 6. However, even if one cannot write a non-constant dimensionless $f(\phi)$, here we can always take $f(\phi) = c$ to be a dimensionless constant, resulting in the allowed counterterm $c \int_{\Sigma^{(3)} \times \mathbb{R}} \text{Tr}(R \wedge R)$. An obvious relation:

$$\int_{\Sigma^{(3)} \times \mathbb{R}} \text{Tr}(R \wedge R) = 192\pi \int_{\mathbb{R}} \text{d}t \frac{\text{d}}{\text{d}t} \int_{\Sigma^{(3)}} \text{CS}_{\text{grav}}, \tag{26}$$

shows that such a counterterm produces a pure gauge Berry connection for the metric, not an actual ambiguity. Therefore, again, only when we can write a dimensionless non-constant $f(\phi)$, the Berry connection involving time-dependent metric becomes truly ambiguous.

At $D = 5$ we find another harmful counterterm:

$$\int_{\Sigma^{(4)} \times \mathbb{R}} \text{Tr}(R \wedge R) \wedge \phi^* \omega, \tag{27}$$

where $\omega$ is an arbitrary one-form, or more generally a gauge field on $M$. Again, the factor of $\phi^* \omega$ absorbs the $\mathbb{R}$ integration, and the above expression factorizes as

$$\left( \int_{\Sigma^{(4)}} \text{Tr}(R \wedge R) \right) \int_{\mathbb{R}} \phi^* \omega = 24\pi^2 \sigma(\Sigma^{(4)}) \int_{\mathbb{R}} \phi^* \omega, \tag{28}$$

where $\sigma(\Sigma^{(4)})$ is the signature of a four-manifold $\Sigma^{(4)}$. Thus this counterterm vanishes for $\Sigma^{(4)}$ of zero signature, (for example, for $S^4$ or $S^2 \times S^2$,) hence removing this sort of ambiguity (of course other observables might still be ambiguous on such curved backgrounds). For $\Sigma^{(4)}$ of non-zero signature, such as $\mathbb{CP}^2$, the counterterm gets activated and renders the Berry connection of $\phi$ unphysical (even if the metric of $\Sigma^{(4)}$ is static).

One can also write a mixed CS term $\int_{\Sigma^{(4)} \times \mathbb{R}} \text{Tr}(R \wedge R) \wedge A$ using the $U(1)$ gauge field $A$, which however contributes a nontrivial inflow for the boundary gauge-gravity anomaly, allowing to fix this ambiguity. The only other term we should worry about in 5D is (17), which may generate another actual ambiguity in the Berry phase.

At $D = 6$ the gravitational counterterms include $\Lambda^4 f(\phi)R$, $\Lambda^2 R^2$, $\Lambda^2 W^2$, $\Lambda^2 E_4$, and the finite counterterms: $R^3$, $E_6$, and the three Weyl anomalies $I_i$ [39]. None of these contribute terms with precisely one time derivative, so they can be ignored. There are also no new gravitational characteristic classes, so the only contributing counterterms activated on curved backgrounds are built from the lower degree characteristic classes:

$$\int_{\Sigma^{(5)} \times \mathbb{R}} \text{Tr}(R \wedge R) \wedge A \wedge \phi^* \tau, \quad [\tau] \in H^1(M, \mathbb{Z}), \tag{29}$$

$$\int_{\Sigma^{(5)} \times \mathbb{R}} f(\phi) \text{Tr}(R \wedge R) \wedge F, \tag{30}$$

where $A$ and $F = \text{d}A$ correspond to a $U(1)$ global symmetry. The cohomology class $[\tau]$ is fixed from the agreement with boundary mixed anomalies, since (29) is a secondary class. The remaining ambiguity $\tau \mapsto \tau + \text{d}f$ again has two effects. If $A$ and the metric are time-independent, it corresponds to gauge transformations of the Berry connection for $\phi$. On the other hand, if $\tau = \text{d}f$, upon integration by parts, (29) becomes the second term (30). For the time-dependent gauge field $A$ and/or metric, (30) makes the corresponding Berry connection

ambiguous. The only other class of counterterms we should care about (for time-dependent background gauge fields) is (19), which again can induce ambiguities.

It is clear how this analysis generalizes to higher dimensions. In $D = 2k + 2$ we have the following two types of counterterms:[7]

$$\int_{\Sigma_{2k+1} \times \mathbb{R}} S_{2k+1}[\Gamma, A] \wedge \phi^* \tau, \quad [\tau] \in H^1(M, \mathbb{Z}),$$
$$\int P_{2k+2}[R, F] f(\phi), \tag{31}$$

where $S_{2k+1}[\Gamma, A]$ represents some secondary invariant built out of the Levi-Civita connection and the background gauge fields $A$, and $P_{2k+2}$ is a characteristic $(2k + 2)$-form built from the Riemann tensor and the background gauge curvatures $F$. Only counterterms of the latter type introduce genuine ambiguities. Likewise, in $D = 2k + 1$ dimension, we expect two analogous types of counterterms:

$$\int_{\Sigma_{2k} \times \mathbb{R}} S_{2k+1}[\Gamma, A],$$
$$\int P_{2k}[R, F] \wedge \phi^* \omega, \quad \omega \in \Omega^1(M), \tag{32}$$

where again only the latter one leads to genuine ambiguities.

## 2.3 tt* connection

Berry connection in QFT has been quite useful in deriving the tt* equations or vacuum geometries in theories with four supercharges. Originally formulated in [23] for 2D $\mathcal{N} = (2, 2)$ theories on a cylinder, they were later generalized to 3D and 4D theories in [25, 27]. The original constructions proceed under the assumption that the Berry connection makes sense in the corresponding QFT setup. Let us briefly look at those constructions and confirm that such an assumption agrees with our earlier analysis of counterterms.

1. In [23] the Berry connection for chiral couplings is considered in $\mathcal{N} = (2, 2)$ theories on $S^1 \times \mathbb{R}$ in the Ramond-Ramond (RR) sector. Such couplings are the background scalar parameters, and there are no background gauge fields activated. This is the simplest case of our analysis above, where the Berry connection is clearly well-defined as a term in the 1D effective action.

2. Another application [25] is for the twisted chiral couplings in $\mathcal{N} = (2, 2)$ theories on $S^1 \times \mathbb{R}$ in the RR sector. The scalar couplings are twisted masses $m$ given by the vevs of scalars in the background vector multiplets for flavor symmetries. Additionally, flavor holonomies $h$ along $S^1$ in the same background multiplets are considered as parameters. The Berry connection for such an extended set of parameters could be ambiguous due to the counterterm $f(m) \text{Tr} F$. Here $f(m)$ must be dimensionless, while the mass dimension of $m$ is one, so $f(m) = \text{const} = \alpha$, and the only finite counterterm is the flavor theta-angle $\alpha \text{Tr} F$, which gives the contribution $\alpha \text{Tr} dh$ to the Berry connection. This makes holonomy (not the curvature!) of the Berry connection ambiguous whenever the flavor group has abelian factors. In many cases, it is nonabelian, thus the theta-term is absent and there are no ambiguities.

---

[7]Let us remind once again that the terms of the kind $S_{2s+1} \wedge \phi^* \tau$, $[\tau] \in H^{2(k-s)+1}(M, \mathbb{Z})$ and $P_{2s+1} \wedge \phi^* \omega$, $\omega \in \Omega^{2(k-s)}(M)$, do exist, however, they are not activated on backgrounds with spatially constant $\phi$.

3. In the generalization to 3D $\mathcal{N} = 2$ theories on $\mathbb{T}^2 \times \mathbb{R}$, [25] studied the Berry connection for real twisted masses $m$ and the corresponding flat flavor connections on $\mathbb{T}^2$. Thus for each unit of rank of the flavor group, we have an $\mathbb{R} \times \check{\mathbb{T}}^2$ worth of parameters. According to our analysis, we should worry about the counterterm $\int_{\mathbb{T}^2 \times \mathbb{R}} F \wedge \phi^* \omega$, where $\omega$ is a one-form built out of masses. This counterterm does not get activated because the flavor connections are flat along $\mathbb{T}^2$, $F|_{\mathbb{T}^2} = 0$, and $\phi^* \omega$ absorbs the integral along $\mathbb{R}$.[8] There are also Chern-Simons terms for flavor symmetries that contribute non-trivially to the Berry connection. Their coefficients are unambiguously fixed to agree with the boundary anomalies.

4. In [25], the generalization to 4D $\mathcal{N} = 1$ theories on $\mathbb{T}^3 \times \mathbb{R}$ was also studied. The parameters include flat flavor connections on $\mathbb{T}^3$. Additionally, for each $U(1)$ factor of the *gauge* group, there is an extra $\check{\mathbb{T}}^3 \times \mathbb{R}$ worth of parameters. Here $\zeta \in \mathbb{R}$ is the FI term, and $\check{\mathbb{T}}^3$ labels flat connections for the topological one-form symmetry on $\mathbb{T}^3$ (whose background gauge field $B_{\mu\nu}$ couples to the dynamical $U(1)$ gauge field via $\int B \wedge F$). Our analysis above did not include higher-form symmetries, but it is straightforward to generalize it. The only potentially harmful counterterms involving ordinary background gauge fields are (19): $\int f(\zeta) \text{Tr}(F \wedge F)$ and $\int f(\zeta) \text{Tr}(F) \wedge \text{Tr}(F)$. They vanish because our background is flat along $\mathbb{T}^3$. The counterterms including $B$ are $\int dB \wedge \phi^* \omega$, where $\omega$ is a one-form built out of the FI terms, and $\int B \wedge F$. The former vanishes because $B$ is flat, and the latter is a nontrivial inflow term for the boundary mixed anomaly (thus the ambiguity can be fixed). We can also build more general counterterms but they are suppressed by the inverse powers of the cutoff $\Lambda$. We see that all the ambiguities are resolved and the Berry connection is well defined.

Therefore, all the cases of vacuum geometry studied in the literature are based on the unambiguous instances of Berry connection (up to a mild ambiguity for abelian factors of the flavor group in 2d). This is not so unexpected, for in all those cases the underlying mathematical structures turn out to be tractable thanks to SUSY. Such things do not happen by accident. Also note that in the above analysis, we ignored the possibility of having several vacua leading to the nonabelian Berry connection, which is usually the case in the tt$^*$ contexts. When this happens, the IR theory can be approximately described as a direct sum of sectors attached to each vacuum, and the above is expected to hold in each sector.

## 2.4 Adiabatic connections

Our previous analysis shows when the Berry connection makes sense as an unambiguous term in the 1D effective action of a higher-dimensional QFT. This does not mean, however, that the original definition [1,2] via projection of the trivial connection immediately applies. Although the Hilbert bundle over the space of parameters is trivial (see [40, page 67] and [41]), there is no canonical trivialization in a general QFT. To choose a trivial connection on the Hilbert bundle, we need to make a choice of trivialization first [3].

A family of QFTs is equipped with a Hilbert bundle

$$\pi : \mathcal{H} \to M \,, \tag{33}$$

over the parameter space $M$, and as emphasized in [3], we need a connection $\nabla^{\mathcal{H}}$ on it to define the projector Berry connection on the gapped subbundle. Generally, there is no canonical choice of $\nabla^{\mathcal{H}}$, and for a generic interacting QFT, it is not even clear how to characterize at least some such connection (although something can be done in the free case, as we will see

---

[8]Even if it was activated, it would be suppressed by $\Lambda^{-1}$ for dimension reasons.

later). Nevertheless, as the effective action considerations show, there exists the adiabatic connection describing response to the slowly varying parameters. Can we characterize it on general grounds?

The authors of [9] take the algebraic approach. In this case one deals with the $C^*$-algebra $\mathcal{A}$ of observables, and a state $\psi$ is a functional $\psi : \mathcal{A} \to \mathbb{C}$ that computes expectation values. The GNS construction connects this to the usual notion of a vev in a pure state in the Hilbert space or in a mixed state described by a density matrix. One can study families of states $\psi_m$ labeled by $m \in M$, and [9] consider smooth families characterized by

$$\mathrm{d}\psi_m(\mathcal{O}) = \psi_m(D\mathcal{O}), \quad D\mathcal{O} = \mathrm{d}\mathcal{O} + [G, \mathcal{O}], \tag{34}$$

where d is the de Rham differential on $M$, and $G$ is some operator-valued one-form on $M$.

The algebraic version of the adiabatic theorem would claim that as the parameters $m(t)$ slowly vary with time, a gapped instantaneous vacuum $\psi^{(0)}_{m(t_0)}$ approximately evolves as $\psi^{(0)}_{m(t)}$. In this context, one can call $G$ the adiabatic connection since

$$\frac{\mathrm{d}}{\mathrm{d}t}\psi_{m(t)}(\mathcal{O}) = \dot{m}^i \psi_{m(t)}(D_i \mathcal{O}) + \psi_{m(t)}\left(\frac{1}{i}[H, \mathcal{O}]\right) = \dot{m}^i \psi_{m(t)}(D_i \mathcal{O}). \tag{35}$$

In [9], the authors essentially use such an algebraic approach to construct the topological term in the effective action on $\mathbb{R}^D$. It also naturally leads to the topological terms on $\mathbb{R}^p$ with $p < D$, when we compactify the theory on $\Sigma^{(D-p)}$. In particular, $G$ itself encodes our Berry connection given by a topological term in the effective action on $\mathbb{R}$.

## 3 Example: Free fermions in 3D

Let us explore an important example of free fermions on $\mathbb{T}^2 \times \mathbb{R}$ coupled to flat background gauge fields on $\mathbb{T}^2$. We choose the periodic spin structure on $\mathbb{T}^2$, although the answer will be easily generalized to other spin structures as well. We also turn on real masses $m$ to keep the theory gapped. Flat gauge fields and masses may slowly vary in time $\mathbb{R}$, as long as the gap does not close. In this background, the counterterm (17) vanishes due to $\mathrm{F}\big|_{\mathbb{T}^2} = 0$,[9] while (20), as usual, is fixed via the inflow argument (although, we will briefly return to (20) in the end of Section 3.1). One could simply integrate out the fermions and look at the effective action: General considerations imply that the effective CS terms will govern the Berry connection. We will look at the problem from this angle in Section 4.

The theory is free, so it is straightforward to construct its Hilbert space as the Fock space, determine the vacuum vector as a function of parameters, and run the standard Berry construction. In this Section we do just that, and in later parts of the paper we compare it with the analysis based on the effective CS terms.

For concreteness, consider a single Dirac fermion of real mass $m$ coupled to the flat $U(1)$ connection $(a_1, a_2)$, so the full set of parameters is $(a_1, a_2, m) \in \mathring{\mathbb{T}}^2 \times \mathbb{R}$:

$$S = \int \mathrm{d}^3 x \left[ \overline{\psi}_-(D_1 + iD_2)\psi_- - \overline{\psi}_+(D_1 - iD_2)\psi_+ + \overline{\psi}_-(\partial_3 + m)\psi_+ + \overline{\psi}_+(\partial_3 - m)\psi_- \right], \tag{36}$$

where $D_j\psi_\pm = (\partial_j + ia_j)\psi_\pm$, $D_j\overline{\psi}_\pm = (\partial_j - ia_j)\overline{\psi}_\pm$. Equations of motion in the Minkowski signature (such that $\partial_0 = i\partial_3$) are:

$$(\partial_0 + im)\psi_+ + (iD_1 - D_2)\psi_- = 0, \qquad (\partial_0 + im)\overline{\psi}_+ + (iD_1 - D_2)\overline{\psi}_- = 0,$$
$$(\partial_0 - im)\psi_- - (iD_1 + D_2)\psi_+ = 0, \qquad (\partial_0 - im)\overline{\psi}_- - (iD_1 + D_2)\overline{\psi}_+ = 0. \tag{37}$$

---

[9]It would be interesting to generalize our analysis to include a non-trivial flux $\mathrm{F}\big|_{\mathbb{T}^2} \neq 0$. It appears to activate (17), however, because $\mathrm{F} \wedge \mathrm{d}m$ has dimension 4, this counterterm is suppressed by the UV cutoff. Something like $\mathrm{F} \wedge \frac{dm}{m}$ would work, but is not a naturally allowed counterterm.

In the momentum space, the canonical anti-commutators are

$$[\psi_-(p), \overline{\psi}_+(q)]_+ = [\psi_+(p), \overline{\psi}_-(q)]_+ = \delta^2(p+q), \tag{38}$$

where $p$, $q$ are spatial momenta. As the space is torus, $p = (p_1, p_2) \in \mathbb{Z}^2$ is discrete, so the anti-commutators of the Fourier modes are more properly written as:

$$[\psi_+(p), \overline{\psi}_-(-q)]_+ = [\psi_-(p), \overline{\psi}_+(-q)]_+ = \delta_{p_1,q_1}\delta_{p_2,q_2}, \tag{39}$$

with the remaining anti-commutators zero. All the Fourier modes together form an infinite dimensional Clifford algebra that we denote as:

$$\mathscr{C} = \mathbb{C}\Big[\{\psi_\pm(p), \overline{\psi}_\pm(-p)|p \in \mathbb{Z}^2\}\Big]/(\text{relations}). \tag{40}$$

The equations of motion (EOM) imply the dispersion relation:

$$E^2(p) = m^2 + \widehat{p}_1^2 + \widehat{p}_2^2, \quad \widehat{p}_i = p_i + a_i, \tag{41}$$

which holds for the quartet of modes $\psi_\pm(p)$ and $\overline{\psi}_\pm(-p)$. The solutions to EOM are:

$$\psi_\pm(p) = \frac{1}{\sqrt{2E(p)}}\left(v_\pm(p)c^+(p)e^{iE_+(p)t} + u_\pm(p)b(p)e^{-iE_+(p)t}\right),$$

$$\overline{\psi}_\mp(-p) = \frac{1}{\sqrt{2E(p)}}\left(u_\pm^*(p)b^+(p)e^{iE_+(p)t} + v_\pm^*(p)c(p)e^{-iE_+(p)t}\right), \tag{42}$$

where

$$v_+(p) = \sqrt{E(p)-m}, \qquad v_-(p) = \frac{i\widehat{p}_1 + \widehat{p}_2}{\sqrt{E(p)-m}},$$

$$u_+(p) = \sqrt{E(p)+m}, \qquad u_-(p) = -\frac{i\widehat{p}_1 + \widehat{p}_2}{\sqrt{E(p)+m}}. \tag{43}$$

Now the anti-commutators are

$$[c^+(p), c(q)]_+ = [b^+(p), b(q)]_+ = \delta_{p_1,q_1}\delta_{p_2,q_2}, \tag{44}$$

and the Dirac's vacuum obeys:

$$b(p)|0\rangle = c(p)|0\rangle = 0, \quad \text{for all } p. \tag{45}$$

## 3.1 Berry phase

Now let us consider how all these structures depend on the parameters $(a_1, a_2, m) \in \check{\mathbb{T}}^2 \times \mathbb{R}$. The algebra of modes $\mathscr{C}$ is clearly trivially fibered over $\mathbb{R}$. As for $\check{\mathbb{T}}^2$, we should take into account that $a_1 \sim a_1 + 1$ and $a_2 \sim a_2 + 1$ are periodic variables, whose periodicity is implemented by the large background gauge transformations $e^{i\varphi_1}$ and $e^{i\varphi_2}$, where $(\varphi_1, \varphi_2)$ are angular coordinates on the spatial torus $\mathbb{T}^2$. For example, the gauge transformation with parameter $e^{i\varphi_1}$ acts as:

$$(a_1 + 1, a_2) \mapsto (a_1, a_2),$$
$$\psi_\pm \mapsto e^{i\varphi_1}\psi_\pm,$$
$$\overline{\psi}_\pm \mapsto e^{-i\varphi_1}\overline{\psi}_\pm. \tag{46}$$

At the level of Fourier modes, this defines a homomorphism of the Clifford algebra:

$$g_1 : \mathscr{C} \to \mathscr{C} \,,$$
$$\psi_\pm(p_1, p_2) \mapsto \psi_\pm(p_1 + 1, p_2) \,,$$
$$\overline{\psi}_\pm(-p_1, -p_2) \mapsto \overline{\psi}_\pm(-p_1 - 1, -p_2) \,, \tag{47}$$

which implements the identification $a_1 \sim a_1 + 1$. Likewise, there is another homomorphism:

$$g_2 : \mathscr{C} \to \mathscr{C} \,,$$
$$\psi_\pm(p_1, p_2) \mapsto \psi_\pm(p_1, p_2 + 1) \,,$$
$$\overline{\psi}_\pm(-p_1, -p_2) \mapsto \overline{\psi}_\pm(-p_1, -p_2 - 1) \,, \tag{48}$$

which implements $a_2 \sim a_2 + 1$. We thus obtain a Clifford bundle on $\mathring{\mathbb{T}}^2$ by starting with a trivial bundle $\mathbb{R}^2 \times \mathscr{C}$, where $(a_1, a_2) \in \mathbb{R}^2$, and identifying the fibers over $(a_1, a_2)$ and $(a_1 + 1, a_2)$ via $g_1$, while the fibers over $(a_1, a_2)$ and $(a_1, a_2 + 1)$ are glued via $g_2$. Let us denote the resulting Clifford bundle by $\widehat{\mathscr{C}}$. Each of its fibers is $\mathscr{C}$, the algebra of fermionic creation and annihilation operators, which admits a representation on the Hilbert space realized as a Fock space. We want to build the corresponding Hilbert bundle $\widehat{\mathcal{H}}$ over $\mathring{\mathbb{T}}^2$.

For each $p \in \mathbb{Z}^2$, we have two canonically conjugate pairs of Grassmann variables, $(\psi_+(p), \overline{\psi}_-(-p))$ and $(\psi_-(p), \overline{\psi}_+(-p))$. They are fixed, independent of the parameters and act on a four-dimensional Fock space:

$$\mathcal{F}_p = \mathbb{C}^4 = \mathrm{Span}\{|\downarrow\downarrow\rangle_p, |\downarrow\uparrow\rangle_p, |\uparrow\downarrow\rangle_p, |\uparrow\uparrow\rangle_p\} \,, \tag{49}$$

defined, for concreteness, via:

$$\psi_\pm(p)|\downarrow\downarrow\rangle_p = 0 \,,$$
$$\overline{\psi}_-(-p)|\downarrow\downarrow\rangle_p = |\uparrow\downarrow\rangle_p \,, \quad \overline{\psi}_+(-p)|\downarrow\downarrow\rangle_p = |\downarrow\uparrow\rangle_p \,,$$
$$\overline{\psi}_-(-p)\overline{\psi}_+(-p)|\downarrow\downarrow\rangle_p = |\uparrow\uparrow\rangle_p \,, \tag{50}$$

with the inner product:

$$_p\langle\downarrow\downarrow | \uparrow\uparrow\rangle_p = {}_p\langle\downarrow\uparrow | \uparrow\downarrow\rangle_p = {}_p\langle\uparrow\downarrow | \downarrow\uparrow\rangle_p = {}_p\langle\uparrow\uparrow | \downarrow\downarrow\rangle_p = 1 \,. \tag{51}$$

The total space of states on $\mathbb{T}^2$ can be defined as[10]

$$\mathbb{V}[\mathbb{T}^2] = \bigotimes_{p \in \mathbb{Z}^2} \mathcal{F}_p \,. \tag{52}$$

This space is too large to be "the Hilbert space" of the theory: It contains a lot of unphysical non-normalizable states. The Hilbert space is identified as (the closure of) the subspace of normalizable states:

$$\mathcal{H} \equiv \mathcal{H}[\mathbb{T}^2] = \overline{\{\psi \in \mathbb{V}[\mathbb{T}^2] : 0 < \langle\psi, \psi\rangle < \infty\}} \,. \tag{53}$$

Let us define the shift operators

$$s_{1,2} : \mathcal{H} \to \mathcal{H} \,, \tag{54}$$

---

[10]Infinite tensor product is the universal object for the multilinear maps from the direct product $\prod_{p \in \mathbb{Z}^2} \mathcal{F}_p$ (which is allowed to be infinite) to $\mathbb{C}$.

which shift either the $p_1$ or the $p_2$ label by one, respectively. More precisely, if we denote the bispinor at momentum $p = (p_1, p_2)$ by $|\sigma_1(p), \sigma_2(p)\rangle_p$, then

$$s_1 \bigotimes_{p \in \mathbb{Z}^2} |\sigma_1(p), \sigma_2(p)\rangle_p = \bigotimes_{p \in \mathbb{Z}^2} e^{i\theta_1(a,p)} |\sigma_1(p), \sigma_2(p)\rangle_{p+\delta_1}, \tag{55}$$

$$s_2 \bigotimes_{p \in \mathbb{Z}^2} |\sigma_1(p), \sigma_2(p)\rangle_p = \bigotimes_{p \in \mathbb{Z}^2} e^{i\theta_2(a,p)} |\sigma_1(p), \sigma_2(p)\rangle_{p+\delta_2}, \tag{56}$$

where $\delta_1 = (1, 0)$, $\delta_2 = (0, 1)$. Here $\theta_1(a, p)$ and $\theta_2(a, p)$ are some ambiguous functions, which we will discuss momentarily. The above definitions of $s_1$ and $s_2$ extend to $\mathcal{H}$ by linearity. The maps of Hilbert spaces $s_{1,2}$ should be compatible with the maps of Clifford algebras $g_{1,2}$ in the sense that the following diagrams commute:

$$
\begin{array}{ccc}
\mathscr{C} & \xrightarrow{\rho} & \mathrm{End}(\mathcal{H}) \\
{\scriptstyle g_1}\downarrow & & \uparrow{\scriptstyle \mathrm{ad}_{s_1}} \\
\mathscr{C} & \xrightarrow{\rho} & \mathrm{End}(\mathcal{H})
\end{array}
\qquad
\begin{array}{ccc}
\mathscr{C} & \xrightarrow{\rho} & \mathrm{End}(\mathcal{H}) \\
{\scriptstyle g_2}\downarrow & & \uparrow{\scriptstyle \mathrm{ad}_{s_2}} \\
\mathscr{C} & \xrightarrow{\rho} & \mathrm{End}(\mathcal{H})
\end{array}
$$

where $\rho$ is the representation morphism, and for $M \in \mathrm{End}(\mathcal{H})$, $\mathrm{ad}_{s_1} M = s_1^{-1} M s_1$ and $\mathrm{ad}_{s_2} M = s_2^{-1} M s_2$. These diagrams say that $g_1$ and $g_2$ are compatible with the adjoint action of $s_1$ and $s_2$, respectively. Note that the unknown $\theta_{1,2}(a, p)$ drop out of these conditions. We next use $s_1$ and $s_2$ as the gluing cocycles to build the Hilbert bundle $\widehat{\mathcal{H}}$ over $\check{\mathbb{T}}^2$. Hence the only other condition is that $\theta_{1,2}(a, p)$ are compatible with the bundle structure (there are no monodromy defects):

$$e^{i\theta_1(a+\delta_1+\delta_2, p) + i\theta_2(a+\delta_2, p+\delta_1)} = e^{i\theta_2(a+\delta_1+\delta_2, p) + i\theta_1(a+\delta_1, p+\delta_2)}. \tag{57}$$

What is the remaining ambiguity in choosing $\theta_{1,2}(a, p)$? It corresponds to the possibility of having background CS terms for the $U(1)$ symmetry coupled to our spinor. Such CS terms could be generated by some extra-heavy fermions at the UV scale. They are completely decoupled from the physics of $\psi, \overline{\psi}$, which is why the Clifford bundle $\widehat{\mathscr{C}}$, the EOMs, the spectrum do not depend on them. However, the global structure of the Hilbert bundle over $\check{\mathbb{T}}^2$ is sensitive to the presence of background CS terms, possibly generated by the UV physics. This is why we get the ambiguity in constructing $\widehat{\mathcal{H}}$.

The minimal choice is of course $\theta_1(a, p) = \theta_2(a, p) = 0$, which corresponds to no background CS level at all. In this case the Hilbert bundle $\widehat{\mathcal{H}}$ admits a trivial connection d, as in the standard Berry connection setup. The annihilation operators depend on the parameters $(m, a_i)$, and up to proportionality factors are given by:

$$b(p) \propto (\widehat{p}_2 - i\widehat{p}_1)\psi_-(p) - (E(p) + m)\psi_+(p), \quad c(p) \propto (\widehat{p}_2 - i\widehat{p}_1)\overline{\psi}_-(-p) + (E(p) + m)\overline{\psi}_+(-p). \tag{58}$$

We can thus identify the vector $|\Omega_p(a)\rangle \in \mathcal{F}_p$ annihilated by $b(p)$ and $c(p)$:

$$|\Omega_p(a)\rangle = \frac{(\widehat{p}_2 - i\widehat{p}_1)| \uparrow\downarrow\rangle_p + (E(p) + m)| \downarrow\uparrow\rangle_p}{\sqrt{(\widehat{p}_2^2 + \widehat{p}_1^2) + (E(p) + m)^2}}, \quad m > 0, \tag{59}$$

which is smooth in the variable $a = a_1 + i a_2$ for $m > 0$. For $m < 0$ it is not, so we scale away the bad factor to obtain a better expression:

$$|\Omega_p(a)\rangle = \frac{(E(p) - m)| \uparrow\downarrow\rangle_p + (\widehat{p}_2 + i\widehat{p}_1)| \downarrow\uparrow\rangle_p}{\sqrt{(\widehat{p}_2^2 + \widehat{p}_1^2) + (E(p) - m)^2}}, \quad m < 0, \tag{60}$$

which is smooth in $a$ for $m < 0$.

The state $|\Omega_p(a)\rangle$ depends on $(m, a_i)$ and undergoes a Bogolyubov rotation as $(m, a_i)$ vary. The total physical vacuum is:

$$|\Omega(a)\rangle = \bigotimes_{p \in \mathbb{Z}^2} |\Omega_p(a)\rangle. \tag{61}$$

The trivial connection d on $\widehat{\mathcal{H}}$ then induces the Berry connection. It is trivial for $m$:

$$A_m = \langle \Omega(a)| \frac{\partial}{\partial m} |\Omega(a)\rangle = 0. \tag{62}$$

For flat gauge fields, it appears to be non-trivial and is contributed by each mode:

$$A_i = \langle \Omega(a)| \frac{\partial}{\partial a_i} |\Omega(a)\rangle = \sum_{p \in \mathbb{Z}^2} A_i^{(p)}, \tag{63}$$

where $A_i^{(p)} = \langle \Omega_p(a)| \frac{\partial}{\partial a_i} |\Omega_p(a)\rangle$, computed for both signs of $m$ using (59) and (60), is:

$$A_1^{(p)} = i \frac{m}{|m|} \frac{\widehat{p}_2}{2(\widehat{p}_1^2 + \widehat{p}_2^2)} \left[ \frac{|m|}{\sqrt{m^2 + \widehat{p}_1^2 + \widehat{p}_2^2}} - 1 \right],$$

$$A_2^{(p)} = -i \frac{m}{|m|} \frac{\widehat{p}_1}{2(\widehat{p}_1^2 + \widehat{p}_2^2)} \left[ \frac{|m|}{\sqrt{m^2 + \widehat{p}_1^2 + \widehat{p}_2^2}} - 1 \right]. \tag{64}$$

One can easily recognize these as gauge potentials for the singular Dirac monopole of magnetic charge one sitting at the location $(-p_1, -p_2, 0)$ in the space parameterized by $(a_1, a_2, m)$. These formulae are written in the patches $m > 0$ and $m < 0$ precisely in such a way that we never encounter the Dirac string.

After the summation over $p \in \mathbb{Z}^2$, the total Berry connection thus describes a doubly-periodic Dirac monopole of charge 1 on $\check{\mathbb{T}}^2 \times \mathbb{R}$. This object is slightly problematic because on the left and on the right from the monopole, the flux through $\check{\mathbb{T}}^2$ (i.e., the Chern number,) is precisely $\pm\frac{1}{2}$, which indicates that the bundle over $\check{\mathbb{T}}^2$ cannot be smooth.[11] This can be checked explicitly for our connection, since the curvature is:

$$F_{12} = \partial_1 A_2 - \partial_2 A_1 = im \sum_{(p_1, p_2) \in \mathbb{Z}^2} \frac{1}{2\left(m^2 + (p_1 + a_1)^2 + (p_2 + a_2)^2\right)^{\frac{3}{2}}}, \tag{65}$$

and we can compute the Chern class:

$$\frac{1}{2\pi i} \int_{\check{\mathbb{T}}^2} F = \frac{m}{2\pi} \int_{\mathbb{R}^2} \frac{dk_1 dk_2}{2\left(m^2 + k_1^2 + k_2^2\right)^{\frac{3}{2}}} = \frac{m}{2|m|}. \tag{66}$$

This is of course related to the known parity anomaly [34–36]: The Dirac fermion of charge 1 and real mass $m$ generates the CS level $\frac{1}{2}\mathrm{sign}(m)$ in the IR, which is understood via the $\eta$-invariant in [42], see also [43]. The half-integral level means that the large gauge transformation responsible for the periodicity $a_1 \sim a_1 + 1, a_2 \sim a_2 + 1$ is in fact *broken*. Instead, the true periodicity of the holonomies is:

$$a_1 \sim a_1 + 2, \quad a_2 \sim a_2 + 2, \tag{67}$$

---

[11]Note that the authors of [25] have also run into this issue in the tt* geometry of a free chiral multiplet, which coincides with the Berry connection of a free Dirac spinor that we are looking at.

and the vacuum bundle should be considered over the torus $\widetilde{\mathbb{T}}^2$ labeled by such $(a_1, a_2)$, which is the four-fold covering of the original $\check{\mathbb{T}}^2$. The magnetic flux through $\widetilde{\mathbb{T}}^2$ then becomes $\pm 2$, and the Berry connection is described by the doubly-periodic Dirac monopole of magnetic charge 4 on $\widetilde{\mathbb{T}}^2 \times \mathbb{R}$. This subtlety disappears if the spinor $\psi$ has charge 2, in which case we would get the monopole of magnetic charge 4 already on $\check{\mathbb{T}}^2 \times \mathbb{R}$.

It is also straightforward to write the gauge potential for $|m| \to \infty$. This limit means that we are infinitely far away from the monopoles located at $m = 0$ and $a \in \mathbb{Z} + i\mathbb{Z}$. From that far, they look like a magnetically charged wall at $m = 0$, and the magnetic field is almost uniform and constant. The appropriate potential (the Berry connection itself) is then:

$$A = i\frac{\pi}{2}\frac{m}{|m|}(a_1 \mathrm{d}a_2 - a_2 \mathrm{d}a_1). \tag{68}$$

More generally, for spin structure $(s_1, s_2)$ on the spatial $\mathbb{T}^2$, where $s_i = 0$ for periodic and $s_i = 1$ for anti-periodic, the answer would match the expectation from abelian spin CS [44]:

$$A = i\frac{\pi}{2}\frac{m}{|m|}(a_1 \mathrm{d}a_2 - a_2 \mathrm{d}a_1 + s_1 \mathrm{d}a_2 + s_2 \mathrm{d}a_1). \tag{69}$$

Let us briefly address the important case of massless fermion, $m = 0$. Without the real mass, at the lattice points $a \in \mathbb{Z} + i\mathbb{Z}$ the gap closes and the Berry connection is ill defined. These points of course corresponds to a single point $\mu \in \check{\mathbb{T}}^2 \times \{0\} \subset \check{\mathbb{T}}^2 \times \mathbb{R}$, which is the location of the monopole. Away from this point along the torus $(\check{\mathbb{T}}^2 \setminus \mu) \times \{0\}$, the Berry curvature is clearly zero. This is not only seen from (65) at $m = 0$, but also obvious physically: The lattice of monopoles sits along plane $m = 0$, thus the magnetic field at this plane (away from the monopoles) is parallel to it. This means that the pull-back of curvature $F_{ij}$ to the plane vanishes. What about the holonomies along the one-cycles of $\check{\mathbb{T}}^2 \setminus \mu$? If we write a monopole of charge $b$ in the angular coordinates as $A = \frac{b}{2}(1 - \cos\theta)\mathrm{d}\varphi$, we can orient them in such a way that the plane $m = 0$ corresponds to $\theta = \frac{\pi}{2}$. Then the monopole potential becomes simply $\frac{b}{2}\mathrm{d}\varphi$ alogn the plane. For concreteness, we take $b$ even (to avoid issues with the parity anomaly). Then the total potential sourced by the lattice becomes:

$$A = \frac{b}{2}\sum_{n,m \in \mathbb{Z}} \mathrm{d}\varphi_{n,m}, \tag{70}$$

where at a point $a \in \mathbb{C}$, we define $\varphi_{n,m} = \arg(a - n - im)$. This can be carefully regularized to show that there are no holonomies, at least when $b$ is even. We therefore conclude that the Berry connection is trivial on $\check{\mathbb{T}}^2 \setminus \mu$ at $m = 0$. It is ill defined at the point $\mu$, however, if we only focus on the $m = 0$ case, (not the entire parameter space $\check{\mathbb{T}}^2 \times \mathbb{R}$ carrying the monopole-like connection,) it makes sense to extend the trivial connection over the point $\mu$. Note also that for the non-zero mass, the Berry connection was proportional to $\frac{m}{|m|} = \text{sign}(m)$, thus the useful mnemonic to include the trivial connection at $m = 0$ is to define $\text{sign}(0) = 0$.

Finally, if we add a background CS level $k$ (which is a counterterm of the type (20)), the phases $\theta_{1,2}(a, p)$ cannot be zero any more. As a result, the trivial connection $\nabla = \mathrm{d}$ is not define. Instead we have $\nabla = \mathrm{d} + \Theta$ with some local one-form $\Theta$. The shortcut to determining $\Theta$ is the following observation. The only effect of the background CS level $k$ is to replace the Hilbert bundle $\widehat{\mathcal{H}}$ by $\widehat{\mathcal{H}} \otimes \mathcal{L}^k$, where $\mathcal{L}^k$ is the line bundle of Chern class $k$ on $\check{\mathbb{T}}^2 \times \mathbb{R}$. The CS term equips $\mathcal{L}^k$ with the connection determined by the local one-form $ik\pi(a_1 \mathrm{d}a_2 - a_2 \mathrm{d}a_1)$ (see Section 4). This is precisely our $\Theta$, which thus clearly shifts the Berry connection, adding $k$ units of flux.

## 3.2 Boundary states

Let us also explore the dependence of boundary states on the parameters $(m, a_i)$, which turns out to be a closely related question. Namely, for a Euclidean QFT on $\mathbb{T}^2 \times \mathbb{R}$, we cut $\mathbb{R}$ and

impose local elliptic boundary conditions along the boundary $\mathbb{T}^2$. Boundary states are defined as special vectors in $\mathbb{V}[\mathbb{T}^2]$ (but not in $\mathcal{H}[\mathbb{T}^2]$) that, for all practical purposes of computing partition functions and correlators, imitate the effect of boundary conditions.

Consider two types of chiral boundary conditions, which eliminate either the right-moving or the left-moving components of fermions along the boundary:

$$
\begin{aligned}
B_+ : \quad & \psi_+\big| = \overline{\psi}_+\big| = 0\,, \\
B_- : \quad & \psi_-\big| = \overline{\psi}_-\big| = 0\,.
\end{aligned}
\tag{71}
$$

Both are elliptic boundary conditions. Their corresponding boundary states are defined as

$$
\begin{aligned}
|B_+\rangle = \bigotimes_p \sqrt{2}|\downarrow\uparrow\rangle_p\,, \qquad & \langle B_+| = \bigotimes_p \sqrt{2}\langle\downarrow\uparrow|_p\,, \\
|B_-\rangle = \bigotimes_p \sqrt{2}|\uparrow\downarrow\rangle_p\,, \qquad & \langle B_-| = \bigotimes_p \sqrt{2}\langle\uparrow\downarrow|_p\,,
\end{aligned}
\tag{72}
$$

where vectors and covectors represent the right and the left boundaries, respectively.

The somewhat arbitrary-looking coefficients $\sqrt{2}$, on the one hand, ensure the key property of boundary states: Their overlaps, such as $\langle B_+|e^{-TH}|B_+\rangle$, compute the interval partition functions. Indeed, a calculation shows that

$$
\langle B_+|e^{-TH}|B_+\rangle = \prod_p \frac{\widehat{p}_2 - i\widehat{p}_1}{E(p)}(1 - e^{-2TE(p)})\,,
\tag{73}
$$

which can be checked to agree with the $B_+B_+$ interval partition function (assuming we subtract the zero point energy), and similarly for other pairs of boundary conditions.

On the other hand, $|B_\pm\rangle$ are manifestly not normalizable, they belong to $\mathbb{V}[\mathbb{T}^2]$ but not $\mathcal{H}[\mathbb{T}^2]$. The Euclidean evolution for arbitrarily short time $\epsilon > 0$, however, makes it normalizable. Indeed, we can compute the norm,

$$
|e^{-\epsilon H}|B_+\rangle|^2 = \prod_p \frac{E(p) + m + (E(p) - m)e^{-4\epsilon E(p)}}{E(p)}\,,
\tag{74}
$$

which is convergent for any $\epsilon > 0$ but not at $\epsilon = 0$. The boundary states have finite overlaps with physical states. In particular, the overlaps with the vacuum (59) for $m > 0$ are:

$$
\langle\Omega(a)|B_+\rangle = \prod_{p\in\mathbb{Z}^2} \sqrt{\frac{E(p) + m}{E(p)}}\,, \qquad \langle B_+|\Omega(a)\rangle = \prod_{p\in\mathbb{Z}^2} \frac{\widehat{p}_2 - i\widehat{p}_1}{\sqrt{E(p)(E(p) + m)}}\,,
\tag{75}
$$

while with the vacuum (60) written for $m < 0$, the overlaps are:

$$
\langle\Omega(a)|B_+\rangle = \prod_{p\in\mathbb{Z}^2} \frac{\widehat{p}_2 - i\widehat{p}_1}{\sqrt{E(p)(E(p) - m)}}\,, \qquad \langle B_+|\Omega(a)\rangle = \prod_{p\in\mathbb{Z}^2} \sqrt{\frac{E(p) - m}{E(p)}}\,.
\tag{76}
$$

Recall from the analysis of Berry connection that for $m > 0$, $|\Omega(a)\rangle$ is a section of some line bundle $\mathcal{L}^{\frac{1}{2}}$ over $\mathring{\mathbb{T}}^2$, while $\langle\Omega(a)|$ is a section of $\mathcal{L}^{-\frac{1}{2}}$. For $m < 0$, the bundles are swapped, and $|\Omega(a)\rangle$ is a section of $\mathcal{L}^{-\frac{1}{2}}$. Now notice that $\langle\Omega(a)|B_+\rangle$ in (75) is a global function of $a$, not a section, precisely when $m > 0$, while for $m < 0$, $\langle B_+|\Omega(a)\rangle$ in (76) is globally defined. Thus $|B_+\rangle$ and $\langle B_+|$ must be, in some sense, sections of the same bundle $\mathcal{L}^{\frac{1}{2}}$:

$$
|B_+\rangle, \langle B_+| \text{ `` }\in\text{ '' } \Gamma\left(\mathring{\mathbb{T}}^2, \mathcal{L}^{\frac{1}{2}}\right)\,.
\tag{77}
$$

At the same time, $\langle B_+|\Omega(a)\rangle$ in (75) and $\langle\Omega(a)|B_+\rangle$ in (76) must be sections of $\mathcal{L}^{\frac{1}{2}}\otimes\mathcal{L}^{\frac{1}{2}}$. Up to a globally defined function, these sections behave as $\prod_p(\widehat{p}_1+i\widehat{p}_2)$. This is a well known infinite product that is regularized to a theta-function $\vartheta(a)$, where $a=a_1+ia_2$, defined as

$$\vartheta(a)=(x^{1/2}-x^{-1/2})\prod_{n>0}(1-q^n x)(1-q^n/x), \quad \text{where } x=e^{2\pi ia}, \tag{78}$$

and $q$ determines the complex structure of the spatial torus $\mathbb{T}^2$ (in our conventions $q=e^{-2\pi}$). Thus the bundle $\mathcal{L}^{\frac{1}{2}}\otimes\mathcal{L}^{\frac{1}{2}}=\mathcal{L}$ is characterized by the same factor of automorphy as

$$\vartheta(a). \tag{79}$$

The square root on $\mathcal{L}^{\frac{1}{2}}$ is of course an artifact of the parity anomaly.

Equation (77) clearly reflects the boundary 't Hooft anomaly carried by $B_+$. In the absence of bare Chern-Simons terms, the left and right boundary conditions $B_+$ carry the same boundary anomaly $\mathcal{A}$, which is why we see $\mathcal{L}^{\frac{1}{2}}$ both for $|B_+\rangle$ and $\langle B_+|$. For $B_-$, the anomaly would be $-\mathcal{A}$ and the bundle would be $\mathcal{L}^{-\frac{1}{2}}$.

Yet, (77) looks very confusing given that the definitions (72) of the boundary states contain no $a$-dependence. In fact, equations (72) rightfully suggest that $|B_\pm\rangle$ are constant sections of a trivial bundle over $\check{\mathbb{T}}^2$. Then what does (77) mean? It turns out that the more precise statements (that can now be written without the quotation marks) are:

$$e^{-\epsilon H}|B_+\rangle\in\Gamma\left(\check{\mathbb{T}}^2,\mathcal{L}^{\frac{1}{2}}\right), \qquad \langle B_+|e^{-\epsilon H}\in\Gamma\left(\check{\mathbb{T}}^2,\mathcal{L}^{\frac{1}{2}}\right), \tag{80}$$

where $\epsilon>0$ is the regularization parameter that makes the boundary state normalizable (the zero point energy is subtracted from $H$). At $\epsilon=0$ we would have constant sections (of a trivial bundle over $\check{\mathbb{T}}^2$ with fibers $\mathbb{V}[\mathbb{T}^2]$) valued outside the Hilbert space. At any $\epsilon>0$, we find sections of a non-trivial line bundle over $\check{\mathbb{T}}^2$ whose fibers lie inside the physical Hilbert space $\mathcal{H}[\mathbb{T}^2]$. This singular behavior of the boundary states is expected to be typical in QFT. If some boundary condition $B$ is independent of the parameters that we vary (like our $a$) and supports a non-trivial anomaly with respect to that parameter, then in fact $|B\rangle$ *must* lie outside the Hilbert space (have infinite norm). If it did not, $e^{-\epsilon H}|B\rangle$ would be a section of the trivial bundle, which is prohibited by the anomaly.[12]

Let us see this more explicitly. Dropping the zero point energy again, the *regularized* boundary state is

$$e^{-\epsilon H}|B_+\rangle=\bigotimes_p\sqrt{2}\frac{(|f_p|^2+e^{-2\epsilon E(p)})|\downarrow\uparrow\rangle_p+f_p(1-e^{-2\epsilon E(p)})|\uparrow\downarrow\rangle_p}{|f_p|^2+1}, \quad \text{where } f_p=\frac{\widehat{p}_2-i\widehat{p}_1}{E(p)-m}. \tag{81}$$

The norm of this state is finite (74) but not unit. We thus normalize it:

$$|NB_+\rangle=\bigotimes_p\frac{(|f_p|^2+e^{-2\epsilon E(p)})|\downarrow\uparrow\rangle_p+f_p(1-e^{-2\epsilon E(p)})|\uparrow\downarrow\rangle_p}{\sqrt{|f_p|^2+1}\sqrt{|f_p|^2+e^{-4\epsilon E(p)}}}. \tag{82}$$

The projector connection for the line bundle spanned by $|NB_+\rangle$ is

$$B_i=\langle NB_+|\frac{\partial}{\partial a_i}|NB_+\rangle=\sum_p B_i^{(p)}, \tag{83}$$

---

[12]That the boundary 't Hooft anomalies demand $e^{-\epsilon H}|B\rangle$ to be a section of the nontrivial bundle can be seen from the interval partition function. Indeed, after the interval reduction, the boundary anomalies become ordinary 't Hooft anomalies of the 2d theory. As is well-known, they imply that the partition function is a section of a nontrivial bundle over the space of background connections (flat connections in our case). Viewing the interval partition function as the overlap of regularized boundary states $e^{-\frac{T}{2}H}|B\rangle$ and $\langle B|e^{-\frac{T}{2}H}$, we immediately conclude that they should also be sections of the nontrivial bundles.

where $B_i^{(p)}$ is a contribution from the mode $p$ in (82). The answer strongly depends on $\epsilon$. For $\epsilon = 0$, (82) becomes $|NB_+\rangle = \otimes_p |\downarrow\uparrow\rangle_p$, so clearly $B_i = 0$. In the opposite $\epsilon \to +\infty$ limit, the answer coincides with the vacuum Berry connection. In particular, the curvature:

$$\partial_1 B_2 - \partial_2 B_1 = \sum_p \frac{im}{2(m^2 + (p_1 + a_1)^2 + (p_2 + a_2)^2)^{3/2}}, \tag{84}$$

which has the Chern number $\frac{1}{2}\frac{m}{|m|}$, as we have seen before.

Interestingly, the answer is different for finite $\epsilon$. The curvature is $\sum_p F_{12}^{(p)}$, where

$$F_{12}^{(p)} = \partial_1 S_p \partial_2 \overline{S}_p - \partial_2 S_p \partial_1 \overline{S}_p, \tag{85}$$

and

$$S_p = \frac{\overline{f}_p (1 - e^{-2\epsilon E(p)})}{\sqrt{|f_p|^2 + 1}\sqrt{|f_p|^2 + e^{-4\epsilon E(p)}}}. \tag{86}$$

We next integrate it over the base to find the Chern class:

$$\int_{\check{\mathbb{T}}^2} \sum_p F_{12}^{(p)} da_1 da_2 = \int_{\mathbb{R}^2} F_{12}^{(p=0)} da_1 da_2 = \int_{\mathbb{C}} dS_0 \wedge d\overline{S}_0. \tag{87}$$

If we write $a_1 + ia_2 = \rho e^{i\varphi}$, then one can see that at infinity in $\mathbb{C}$, asymptotically $S_0 \to ie^{-i\varphi}/\sqrt{2}$ and $\overline{S}_0 \to -ie^{i\varphi}/\sqrt{2}$. Thus, applying the Stokes theorem, the integral evaluates to

$$\int_{S_\infty^1} S_0 d\overline{S}_0 = \frac{i}{2} \int d\varphi = i\pi, \tag{88}$$

and the Chern number is $\frac{1}{2\pi i}\int F = \frac{1}{2}$.

We thus find that the Chern class $c_1(B_+)$ of the bundle that $e^{-\epsilon H}|B_+\rangle$ is valued in depends discontinuously on $\epsilon$:

$$c_1(B_+) = \begin{cases} 0, & \epsilon = 0, \\ \frac{1}{2}, & 0 < \epsilon < \infty, \\ \frac{m}{2|m|}, & \epsilon = \infty. \end{cases} \tag{89}$$

This counter-intuitive property, on the practical level, is traced back to $\lim_{\epsilon\to\infty} S_0$ being singular at $a_1 = a_2 = 0$ precisely for $m < 0$. This makes the Stokes theorem inapplicable in such a limit, explaining why the finite $\epsilon$ and $\epsilon = \infty$ answers differ for $m < 0$.

The $\epsilon = 0$ case is unphysical (boundary state is not in the Hilbert space). The $\epsilon = \infty$ case just captures the Berry curvature of the vacuum. The case of $0 < \epsilon < \infty$ is the most interesting one: the regularized boundary state $e^{-\epsilon H}|B_+\rangle$ spans the line bundle of Chern class $c(B_+) = \frac{1}{2}$ that captures the boundary anomaly!

## 3.3 SUSY extension

Let us also consider the $\mathcal{N} = 2$ extension, i.e., enrich the story by adding a complex scalar $\phi$ of real mass $m$, such that together with fermions we have a 3d $\mathcal{N} = 2$ chiral multiplet $\Phi$ of real mass $m$. In this case, the zero point energy automatically vanishes, so we do not need to subtract it by hands in various formulas. This case is also of direct relevance for our main applications discussed in Section 5.

We use the same boundary conditions on fermions and SUSY-complete them into the $(0,2)$ boundary conditions:

$$\begin{aligned} B_+ : & \quad \phi\big| = 0, & \psi_+\big| = \overline{\psi}_+\big| = 0, \\ B_- : & \quad \partial_\perp \phi\big| = 0, & \psi_-\big| = \overline{\psi}_-\big| = 0. \end{aligned} \tag{90}$$

Quantization of the complex scalar field gives:

$$\phi = \frac{1}{2\pi}\sum_{p\in\mathbb{Z}^2}\frac{1}{\sqrt{2E(p)}}\left(a_p^+ e^{ipx+iE(p)t} + \overline{a}_p e^{ipx-iE(p)t}\right),$$

$$\overline{\phi} = \frac{1}{2\pi}\sum_{p\in\mathbb{Z}^2}\frac{1}{\sqrt{2E(p)}}\left(\overline{a}_p^+ e^{-ipx+iE(p)t} + a_p e^{-ipx-iE(p)t}\right), \tag{91}$$

where $E(p)$ is the same as in (41), and the commutators of creation/annihilation operators for particles and anti-particles, respectively, are:

$$[a_p, a_q^+] = [\overline{a}_p, \overline{a}_q^+] = \delta_{p,q}. \tag{92}$$

The bosonic Fock vacuum is defined as

$$a_p|\Omega\rangle_b = \overline{a}_p|\Omega\rangle_b = 0. \tag{93}$$

The boundary state $|B_+\rangle = |B_+\rangle_f |B_+\rangle_b$ has the fermionic part from the previous subsection and the following bosonic part:

$$|B_+\rangle_b = N e^{-\sum_p a_p^+ \overline{a}_p^+}|\Omega\rangle_b, \tag{94}$$

which obeys the required condition

$$\phi(t=0)|B_+\rangle_b = \overline{\phi}(t=0)|B_+\rangle_b = 0. \tag{95}$$

This $|B_+\rangle_b$ is an infinite-norm state, but its Euclidean evolution,

$$e^{-\frac{T}{2}H}|B_+\rangle_b = N e^{-\sum_p e^{-TE(p)} a_p^+ \overline{a}_p^+}|\Omega\rangle_b, \tag{96}$$

is a finite-norm state of the norm:

$$|N|^2 \prod_{p\in\mathbb{Z}^2}\frac{1}{1-e^{-2TE_+(p)}}. \tag{97}$$

The normalization $N$ must be taken as $|N|^2 = \prod_p E(p)$, so that $\langle B_+|_b e^{-TH}|B_+\rangle_b$ agrees with the interval partition function. This $|N|^2$ cancels against the similar factor for the fermions. We also see that the $T$-dependent terms $(1-e^{-2TE_+(p)})$ and $(1-e^{-2TE_+(p)})^{-1}$ cancel between the fermions and the bosons, respectively. Thus the interval partition function of the chiral multiplet agrees with the overlap of Euclidean-evolved boundary states when we include both bosons and fermions:

$$\langle B_+|e^{-TH}|B_+\rangle = \prod_P (\widehat{p}_1 + i\widehat{p}_2) = \mathcal{N}\cdot\vartheta(a), \tag{98}$$

where $\mathcal{N}$ is an infinite constant. This agrees with the 2d $(0,2)$ Fermi multiplet index [45–50], as expected (indeed, the interval zero modes in the case of $B_+$ boundary conditions form a Fermi multiplet, hence the $I\times\mathbb{T}^2$ partition function computes its index).

We can also equally easily write the bosonic part of $|B_-\rangle$:

$$|B_-\rangle_b = e^{\sum_p a_p^+ \overline{a}_p^+}|\Omega\rangle_b, \tag{99}$$

which obeys:

$$\dot{\phi}(t=0)|B_-\rangle_b = \dot{\overline{\phi}}(t=0)|B_-\rangle_b = 0. \tag{100}$$

The norm of $e^{-\frac{T}{2}H}|B_-\rangle_b$ is the same as for $e^{-\frac{T}{2}H}|B_+\rangle_b$, and the rest of analysis goes through.

To compute the possible Berry phase, we also write the creation/annihilation operators in terms of modes $\phi(p)$:

$$a_p = \sqrt{\frac{E(p)}{2}}\left(\overline{\phi}_p + \frac{1}{E(p)}\frac{\partial}{\partial \phi_p}\right), \qquad \overline{a}_p = \sqrt{\frac{E(p)}{2}}\left(\phi_p + \frac{1}{E(p)}\frac{\partial}{\partial \overline{\phi}_p}\right),$$

$$\overline{a}_p^+ = \sqrt{\frac{E(p)}{2}}\left(\overline{\phi}_p - \frac{1}{E(p)}\frac{\partial}{\partial \phi_p}\right), \qquad a_p^+ = \sqrt{\frac{E(p)}{2}}\left(\phi_p - \frac{1}{E(p)}\frac{\partial}{\partial \overline{\phi}_p}\right). \tag{101}$$

Then the normalized vacuum is represented by the wave function:

$$\langle \phi, \overline{\phi} | \Omega \rangle_b = \prod_{p \in \mathbb{Z}^2} \sqrt{\frac{2E(p)}{\pi}} e^{-E(p)\overline{\phi}_p \phi_p}. \tag{102}$$

Its contribution to the Berry connection is trivial:

$$\langle \Omega |_b \frac{\partial}{\partial m} | \Omega \rangle_b = \langle \Omega |_b \frac{\partial}{\partial a_i} | \Omega \rangle_b = 0. \tag{103}$$

This immediately follows from the wave function being real and normalized. In a similar situation in quantum mechanics, a one-line proof of this fact is:

$$\langle \psi | \frac{\partial}{\partial a_i} | \psi \rangle = \int d^n x \, \psi(x) \frac{\partial}{\partial a_i} \psi(x) = \frac{1}{2}\frac{\partial}{\partial a_i} \int d^n x \, \psi(x)^2 = \frac{\partial 1}{\partial a_i} = 0. \tag{104}$$

We could of course make the Berry connection $A_i$ non-zero by multiplying the wave function by a non-constant phase (and hence making it complex), but this would only amount to a gauge transformation of $A_i$.

The regularized and normalized boundary state for bosons is:

$$|NB_+\rangle_b = \prod_{p \in \mathbb{Z}^2} \left(\sqrt{1 - e^{-2\epsilon E(p)}} e^{-e^{-\epsilon E(p)} a_p^+ \overline{a}_p^+}\right) |\Omega\rangle_b. \tag{105}$$

Without much work, we see that the corresponding wave function is real. Indeed, $|\Omega\rangle_b$ is represented by the real wave function, $E(p)$ is real, and the differential operator

$$a_p^+ \overline{a}_p^+ = \frac{E(p)}{2}\left(\phi_p - \frac{1}{E(p)}\frac{\partial}{\partial \overline{\phi}_p}\right)\left(\overline{\phi}_p - \frac{1}{E(p)}\frac{\partial}{\partial \phi_p}\right), \tag{106}$$

is real. Therefore, by the same argument, the projector connection one-form on the line bundle spanned by $|NB_+\rangle_b$ is zero.

We see that bosons do not contribute to the vacuum bundle and connection. This matches our expectation that the Berry connection is captured by the effective CS terms generated by the fermions. Likewise, only fermions contribute to the boundary anomaly and thus to the topology of the bundle of boundary states.

The presence of bosons, however, improves the behavior of partition functions. As we saw, the interval partition function (with SUSY boundary conditions) becomes independent of the interval length. Furthermore, its dependence on parameters simplifies: We find a holomorphic answer $\vartheta(a)$ in the $B_+B_+$ case and a meromorphic answer $\vartheta(a)^{-1}$ in the $B_-B_-$ case (unlike a more complicated non-SUSY partition function (73)).

Also, the Berry connection of a chiral multiplet is the simplest instance of 3d tt$^*$ geometry [25]. It manifestly coincides with the free fermion Berry connection (see footnote 11).

## 3.4 What did we learn?

The main lesson of this section is that the original Berry connection can still be defined and studied in QFT, at least when the counterterm analysis of Section 2 shows that it is unambiguous. On the flip side, the technicalities can be messy. Even for a free Dirac fermion, we had to be careful with the details, encountering, among other things, an ambiguity associated with the background CS level along the way (this ambiguity is easily resolved, as explained in Section 2). There is certainly very little hope to use the original definition via projectors in a general interacting QFT, however. The viewpoint of effective field theory, where we study particular terms in the low energy effective action (those with precisely one time derivative), appears to be more powerful, since we understand effective actions way better than the Hilbert bundle of a family of interacting QFTs. Thus in the remaining sections, we use the effective action approach to the Berry connection.

## 4 General gapped 3D QFT

Suppose we have a family of (trivially) gapped 3D theories that we put on $\Sigma \times S^1$, where both $\Sigma$ and $S^1$ are very large. More specifically, $\Sigma$ should be so large that: (1) any possible curvature couplings do not affect the QFT dynamics and can be neglected; (2) after zooming out (RG-flowing) sufficiently far, such that only the gapped vacuum remains in the spectrum, $\Sigma$ is still of finite size. Additionally, the length $\beta$ of $S^1$ is kept much larger than the energy gap, such that only the vacuum propagating along $S^1$ contributes to the partition function:

$$Z[\Sigma \times S^1] = \mathrm{Tr}_{\mathcal{H}[\Sigma]} U(\beta) = e^{-\theta} + \mathcal{O}(e^{-\#\beta}), \tag{107}$$

where $U(\beta)$ is the evolution operator, $e^{-\theta}$ is the leading vacuum contribution. We were not specific about the boundary conditions along $S^1$, but if the vacuum is unique and bosonic, these are not so important. We can always assume the periodic boundary conditions (which is especially useful if we have SUSY to preserve).

When we do not vary parameters with time (along $S^1$), the leading contribution is simply due to the vacuum energy. It is called the *dynamical phase* $e^{-iE_0 t}$, and in Euclidean signature, $it = \beta$, it becomes a real number $e^{-\beta E_0}$ often referred to as the Casimir factor. When we vary parameters adiabatically along $S^1$ (without closing the gap), we find an additional contribution to $\theta$ – the *geometric* or Berry phase [1, 2, 4], which is *phase* even in the Euclidean signature. In SUSY theories $E_0 = 0$, so the Berry phase is the only leading contribution.

As we go around $S^1$, we can let the theory trace out any loop in the space of parameters. Thus $Z[\Sigma \times S^1]$ will capture the Berry phase along such a loop, and in this way, at least in principle, we can determine holonomies of the Berry connection along all possible loops. They capture the complete data of the gauge equivalence class of a connection. In such an approach, we do not need to think about the Hilbert spaces or projector connections. We only have a fiber $\mathbb{C}$ of the vacuum bundle over a fixed point in the parameter space, (which corresponds, say, to the moment of time $0 \in S^1$,) and we have holonomies along all possible loops that start and end at this point. These data fully determine the bundle and the connection.

The computation can be formulated in the language of effective field theory, as explained in the Introduction. We promote parameters to background fields $\phi$ that slowly vary along $S^1$. The partition function is

$$Z[\Sigma \times S^1] = e^{-\Gamma[\phi]}, \tag{108}$$

where $\Gamma[\phi]$ is the effective action. The Berry connection is captured by terms in $\Gamma[\phi]$ with precisely one time derivative.

When $\Sigma$ is very large, $\Gamma[\phi]$ is the 3D action of the invertible field theory that captures response of the gapped vacuum to the variations of parameters $\phi$. We compactify this 3D *classical* action on $\Sigma$, and in the resulting 1D action identify the one-derivative terms.

If the only parameters we consider are background gauge fields, then the important term in $\Gamma$ is the effective Chern-Simons coupling at the given gapped vacuum. This is precisely the term that captures the geometric phase in such a case. Assuming that a given vacuum only preserves the maximal torus $T = U(1)^r$ of the global symmetry group, we thus write the effective abelian CS term:

$$\theta_{\text{Berry}} = \frac{1}{4\pi} \int_{\Sigma \times S^1} k_{ij} A^i \mathrm{d}A^j \,. \tag{109}$$

If we assume that $A^i$ is some abelian connection on the surface $\Sigma$ that slowly changes along the time direction $S^1$, the above can be written as:

$$\theta_{\text{Berry}} = \int_\gamma \mathcal{B} \,, \tag{110}$$

where $\gamma \subset \mathcal{A}[\Sigma]$ is a loop in the space $\mathcal{A}[\Sigma]$ of $U(1)^r$-connections on $\Sigma$, and $\mathcal{B}$ is the Berry connection (local) one-form on this space:

$$\mathcal{B} = \frac{1}{4\pi} \int_\Sigma k_{ij} A^i \delta A^j \,. \tag{111}$$

Here we keep the wedge product implicit, and $\delta$ denotes the de Rham differential on $\mathcal{A}[\Sigma]$. The curvature is

$$\mathcal{F} = \delta\mathcal{B} = \frac{1}{4\pi} \int_\Sigma k_{ij} \delta A^i \delta A^j \,. \tag{112}$$

## 4.1 Flat connections

The above constructions become cleaner if we consider flat $T = U(1)^r$ connections on $\Sigma$ as the parameters, in which case the space of parameters will be denoted as:

$$\mathcal{E}_T[\Sigma] = \text{Hom}(\pi_1(\Sigma), T) \cong \left[ H^1(\Sigma, \mathbb{R}) / H^1(\Sigma, \mathbb{Z}) \right]^r \,. \tag{113}$$

Note that $\mathcal{E}_T[\Sigma] \cong J(\Sigma)^r$, where $J(\Sigma)$ is the Jacobian of $\Sigma$. In case $\Sigma$ is a torus $\mathbb{T}^2$, $J(\Sigma) = \check{\mathbb{T}}^2$, and $\mathcal{E}_T(\mathbb{T}^2) = (\check{\mathbb{T}}^2)^r$ will be simply denoted $\mathcal{E}_T$.

**Genus one.** The case of $\Sigma = \mathbb{T}^2$ is the most straightforward and interesting to us: Indeed, it was analyzed in detail for free theories in Section 3. The space of flat connections $\mathcal{E}_T$ is parameterized by the holonomies $2\pi a_1^i = \int_A A^i$ and $2\pi a_2^i = \int_B A^i$ along the A and B cycles of $\mathbb{T}^2$, which are periodic variables:

$$a_1^i \sim a_1^i + 1 \,, \quad a_2^i \sim a_2^i + 1 \,. \tag{114}$$

The Berry connection becomes:

$$\mathcal{B} = \pi k_{ij} (a_1^i \mathrm{d}a_2^j - a_2^j \mathrm{d}a_1^i) \,, \tag{115}$$

in agreement with the results of Section 3. From the equation (115) we see that as we go around a cycle $a_1^j \mapsto a_1^j + 1$, the connection $\mathcal{B}$ undergoes a gauge transformation by

$$g_1^j = e^{i\pi k_{jr} a_2^r} \,, \tag{116}$$

and likewise $a_2^j \mapsto a_2^j + 1$ corresponds to

$$g_2^j = e^{-i\pi k_{jr} a_1^r}. \tag{117}$$

These transformations are clutching functions of the line bundle over $\mathcal{E}_T$. The composition of transformations associated to $a_1^j \mapsto a_1^j + 1$, $a_2^r \mapsto a_2^r + 1$, $a_1^j \mapsto a_1^j - 1$ and $a_2^r \mapsto a_2^r - 1$ must be trivial, which is the standard smoothness condition for the bundle that here reads as:

$$e^{2\pi i k_{jr}} = 1. \tag{118}$$

Thus the bundle is smooth for $k_{jr} \in \mathbb{Z}$, while for the half-integral $k_{jr}$, we must pass to the torus $\widetilde{\mathbb{T}}^2$ covering $\mathring{\mathbb{T}}^2$ to get a smooth bundle. This is the same effect related to the parity anomaly that was mentioned in Section 3.

The complex structure $\tau$ of $\Sigma = \mathbb{T}^2$ induces one on $\mathcal{E}_T$, with the complex coordinates:

$$x^i = a_1^i + \tau a_2^i. \tag{119}$$

Then the Berry connection and its curvature take the form:

$$\mathcal{B} = \frac{\pi}{\overline{\tau} - \tau} k_{ij}(x^i d\overline{x}^j - \overline{x}^i dx^j), \quad \mathcal{F} = \frac{2\pi}{\overline{\tau} - \tau} k_{ij} dx^i d\overline{x}^j. \tag{120}$$

Since $\mathcal{F}$ has type $(1,1)$ in this complex structure, the $(0,1)$ connection $\overline{\nabla} = \overline{\partial} - i\mathcal{B}^{(0,1)}$ obeys $\overline{\nabla}^2 = 0$ and defines a holomorphic structure on the vacuum line bundle. Holomorphic sections obeying $\overline{\nabla}\psi = 0$ take the form:

$$\psi = e^{-K}\chi(x), \tag{121}$$

where $\chi(x)$ is locally holomorphic and the Kähler potential is

$$K(x,\overline{x}) = \frac{i\pi}{\tau - \overline{\tau}} k_{ij} x^i \overline{x}^j. \tag{122}$$

A global section must obey:

$$\psi(x^j + 1) = g_1^j \psi(x), \qquad \psi(x^j + \tau) = g_2^j \psi(x). \tag{123}$$

It is convenient to redefine $\chi(x) = e^{\frac{i\pi}{\tau - \overline{\tau}} k_{ij} x^i x^j} \Theta(x)$, such that

$$\psi(x,\overline{x}) = e^{\frac{i\pi}{\tau - \overline{\tau}} k_{ij} x^i (x^j - \overline{x}^j)} \Theta(x). \tag{124}$$

Then the property (123) written in terms of $\Theta(x)$ reads:

$$\Theta(x^j + 1) = \Theta(x), \qquad \Theta(x^j + \tau) = e^{-2\pi i k_{jr} x^r - i\pi \tau k_{jj}} \Theta(x). \tag{125}$$

More generally, if $\mu + \nu\tau \in (\mathbb{Z} + \tau\mathbb{Z})^r$ is an arbitrary shift, where $\mu$ and $\nu$ are $r$-component integral vectors, we find:

$$\Theta(x + \mu + \nu\tau) = e^{-2\pi i k(\nu, x) - i\pi \tau k(\nu, \nu)} \Theta(x). \tag{126}$$

The coefficient that appears on the right,

$$\lambda(\mu + \tau\nu, x) = e^{-2\pi i k(\nu, x) - i\pi \tau k(\nu, \nu)}, \tag{127}$$

is the factor of automorphy [51–54] that uniquely fixes the data of a holomorphic line bundle on $\mathcal{E}_T$ (see [55–57]). Given a symmetric matrix of Chern-Simons coefficients $k_{ij} \in \mathbb{Z}$, we can easily construct $\Theta(x)$ obeying (126) using theta functions.

Such a characterization of the holomorphic sections will be important to us very soon, but so far the consideration of holomorphic structure associated to the Berry connection has certainly not been motivated in any way, and is just an exercise. This will change in the next subsection.

**Higher genus.**   Though not of immediate importance to our applications later in this note, let us now briefly consider $\Sigma$ of higher genus $g > 1$. The necessary mathematical background for this subsection can be found in the textbook [58]. Pick one-cycles $(\alpha_s, \beta_s), s = 1..g$ representing a basis in $H_1(\Sigma, \mathbb{Z})$ such that only $\alpha_s$ with $\beta_s$ have a nonzero intersection (equal to one) for all $s = 1..g$. Let one-forms $(\alpha^s, \beta^s)$ represent the dual basis in $H^1(\Sigma, \mathbb{Z})$, which means that $\alpha^s(\alpha_r) = \delta^s_r$, $\beta^s(\beta_r) = \delta^s_r$, while the other contractions vanish. Then the intersection numbers are captured by

$$\int_\Sigma \alpha^s \wedge \beta^t = \delta_{st}, \qquad \int_\Sigma \alpha^s \wedge \alpha^t = \int_\Sigma \beta^s \wedge \beta^t = 0. \tag{128}$$

A general flat $U(1)^r$-connection on $\Sigma$ can be written as:

$$A^i = 2\pi \sum_{s=1}^g (a^i_s \alpha^s + b^i_s \beta^s), \tag{129}$$

subject to the identifications $a^i_s \sim a^i_s + 1$ and $b^i_s \sim b^i_s + 1$. Then the Berry connection is:

$$\mathcal{B} = \sum_{s=1}^g \pi k_{ij}(a^i_s db^j_s - b^i_s da^j_s). \tag{130}$$

Next consider a basis of holomorphic differentials $\omega_s, s = 1..g$ on $\Sigma$, and denote their complex conjugates by $\overline{\omega}_s$. In the decomposition

$$\omega_s = \sum_{r=1}^g \left(\Omega_{s,r} \alpha^r + \Omega_{s,g+r} \beta^r\right), \tag{131}$$

the $g \times 2g$ matrix $\Omega$ is known as the period matrix of $\Sigma$. The period matrix is defined for more general bases of one-forms as well, but for our choice of $(\alpha_s, \beta_s)$, there exists a unique basis of holomorphic differentials $\omega_s$, such that

$$\int_{\alpha_s} \omega_r = \delta_{sr}, \quad 1 \le s, r \le g, \tag{132}$$

and the remaining periods are given by

$$\int_{\beta_s} \omega_r = \tau_{sr}, \tag{133}$$

where $\tau = (\tau_{sr})$ is a symmetric complex $g \times g$ matrix, whose imaginary part is positive definite. By definition, such $\tau$ belongs to the Siegel upper half-space of rank $g$. So, in other words, holomorphic differentials take the form:

$$\omega_s = \alpha^s + \sum_{r=1}^g \tau_{sr} \beta^r. \tag{134}$$

We can define complex coordinates on the space $\mathcal{E}_T[\Sigma] = J(\Sigma)^r$:

$$x^i_s = -b^i_s + \sum_{r=1}^g \tau_{sr} a^i_r, \tag{135}$$

in terms of which the flat abelian gauge field is written as:

$$A^i = 2\pi \sum_{r,s=1}^g \left[(\tau - \overline{\tau})^{-1}\right]_{rs} (x^i_s \overline{\omega}_r - \overline{x}^i_s \omega_r), \tag{136}$$

and the Berry connection becomes:

$$\mathcal{B} = \sum_{r,s=1}^{g} \pi k_{ij} \left[ (\tau - \overline{\tau})^{-1} \right]_{rs} (\overline{x}_s^i dx_r^j - x_s^i d\overline{x}_r^j). \tag{137}$$

Again $\overline{\nabla} = \overline{\partial} - i\mathcal{B}^{(0,1)}$ determines a holomorphic structure on the vacuum line bundle over $J(\Sigma)^r$, and we can write a holomorphic section in the familiar form:

$$\psi = e^{i\pi \sum_{r,s} k_{ij} \left[ (\tau - \overline{\tau})^{-1} \right]_{rs} x_s^i (x_r^j - \overline{x}_r^j)} \Theta(x). \tag{138}$$

Agreement with the clutching functions of the bundle, — that under $a_s^j \mapsto a_s^j + \delta_l^j \delta_s^p$ we perform the gauge transformation by $e^{i\pi k_{li} b_p^i}$, and under $b_s^j \mapsto b_s^j + \delta_l^j \delta_s^p$ we perform the gauge transformation $e^{-i\pi k_{lj} a_p^j}$, — again fixes the periodicity properties of $\Theta$ as

$$\Theta(x_s^i + \delta_l^i \delta_s^p) = \Theta(x_s^i), \quad \Theta(x_s^i + \tau_{sp} \delta_l^i) = e^{-2\pi i k_{lj} x_p^j - i\pi k_{ll} \tau_{pp}} \Theta(x), \tag{139}$$

which clearly and in the most straightforward way generalizes the genus-one case. We can also pick integer-valued vectors $\mu_s^i$, $\nu_s^i$ and write the general formula:

$$\Theta(x + \mu + \tau \nu) = e^{-2\pi i k(\nu,x) - i\pi k(\nu, \tau \nu)} \Theta(x). \tag{140}$$

Again this gives us the factor of automorphy that, for every symmetric matrix of Chern-Simons levels $k_{ij}$, determines the data of a holomorphic line bundle on $J(\Sigma)^r$. One can construct $\Theta(x)$ obeying (140) using the multi-dimensional theta functions.

## 4.2 Holomorphy of partition functions

We found that for flat background gauge fields viewed as parameters, the Berry connection equips the vacuum bundle with the holomorphic structure. We described its holomorphic sections in the previous subsection. One may ask if such sections have any physical significance, – and they indeed do in the presence of SUSY.

This is related to the well-known question of (non)holomorphy of the supersymmetric partition functions, which are in fact sections of non-trivial bundles over the parameter space in the presence of the 't Hooft anomalies [37, 38, 49, 59] (determinant line bundles [60]). One naively expects the partition function to be holomorphic in parameters[13] $x^i$, since $\overline{x}^i$ couple to some $Q$-exact operators, which therefore must decouple [61]. In reality, these $Q$-exact operators fail to decouple in the presence of background gauge fields for symmetries with non-zero 't Hooft anomalies. This is because the $Q$ transformation, in the Wess-Zumino gauge [62], is accompanied by the compensating gauge transformation, including that of the background gauge fields [63]. The latter produces anomalous contributions proportional to the 't Hooft anomalies. Following [38], such a "holomorphic anomaly" of the torus partition function is captured by:

$$\frac{\partial}{\partial \overline{x}^i} \log Z_{\mathbb{T}^2}(x, \overline{x}) = -\frac{i\pi}{\tau - \overline{\tau}} \kappa_{ij} x^j, \tag{141}$$

where $\kappa_{ij}$ is the coefficient of $F^i \wedge F^j$ in the anomaly polynomial. Furthermore, it was argued that the partition function in the scheme that preserves gauge-invariance of $|Z_{\mathbb{T}^2}|^2$ must take the form

$$Z_{\mathbb{T}^2}(x, \overline{x}) = e^{\frac{i\pi}{\tau - \overline{\tau}} \kappa_{ij} x^i (x^j - \overline{x}^j)} \mathcal{I}(x), \tag{142}$$

---

[13]This applies to a wider class of theories, but for us the parameters $x^i$ are flavor fugacities in the elliptic genus of a 2D $\mathcal{N} = (0, 2)$ theory.

where $\mathcal{I}(x)$ is meromorphic and captures the supersymmetric index defined as a super-trace over the Hilbert space. This answer matches our expression (124) for a holomorphic section, provided that $\kappa_{ij} = k_{ij}$, which is not a coincidence, of course. The equation (141) precisely says that the partition function is a holomorphic section of the line bundle over $\mathcal{E}_T \cong (\check{\mathbb{T}}^2)^r$ with respect to the holomorphic structure

$$\overline{\nabla} = \overline{\partial} + \frac{i\pi}{\tau - \overline{\tau}} \kappa_{ij} x^i \mathrm{d}\overline{x}^j \,. \tag{143}$$

The old literature interprets this as a clash between holomorphy and gauge invariance.

Note that (143) coincides with the holomorphic structure $\overline{\partial} - i\mathcal{B}^{(0,1)}$ derived from the Berry connection if $\kappa_{ij} = k_{ij}$. Then the answer (124) or (142) and the periodicity (large gauge transformations) (126) are obtained simply by asking that we have a global meromorphic section written in the unitary gauge/trivialization (the one where $\mathcal{B}^{(0,1)}$ and $\mathcal{B}^{(1,0)}$ are conjugate to each other). The latter, in particular, implies that it acquires a known phase under the large gauge transformations, and so $|Z_{\mathbb{T}^2}|^2$ is gauge invariant.

Thus the holomorphic sections encountered in the previous subsection naturally appear in 2D $\mathcal{N} = (0,2)$ theories as supersymmetric partition functions on $\mathbb{T}^2$, with the matrix $k_{ij}$ capturing the 't Hooft anomalies. In the context of 3D $\mathcal{N} = 2$ theories, the natural ways to obtain similar objects include:

- A state generated by the cigar geometry, i.e., partition function on $D^2 \times_q S^1$, with $D^2$ either having a round hemisphere $HS^2$ background (a half-index) or an A-twisted infinite cigar background (a holomorphic block.[14]) Consider for concreteness the half-index geometry. Along the boundary one imposes 2D $\mathcal{N} = (0,2)$ boundary conditions $\mathcal{B}$ with the boundary anomaly polynomial $\kappa_{ij} F^i F^j$. The same argument that determines the non-holomorphy of the $\mathbb{T}^2$ partition function in 2D also applies to the $HS^2 \times_q S^1$ partition function in 3D, where now it is the 't Hooft anomaly supported at the $\mathbb{T}^2$ boundary that appears in the holomorphic anomaly equation. The boundary torus has $q = e^{-\beta}$, where $\beta$ is the circumference of the $S^1$. As a result, we find

$$Z_{HS^2 \times S^1}(\mathcal{B}) = e^{\frac{i\pi}{\tau - \overline{\tau}} \kappa_{ij} x^i (x^j - \overline{x}^j)} I_{\mathcal{B}}(x) \,, \tag{144}$$

where $I_{\mathcal{B}}(x)$ is the usual operator-counting half-index. The holomorphic (or meromorphic) combination $e^{\frac{i\pi}{\tau - \overline{\tau}} \kappa_{ij} x^i x^j} I_{\mathcal{B}}(x)$ has previously appeared in the literature [49, 65], but we need to include the non-holomorphic piece in (144) to get the answer in the regularization scheme compatible with gauge transformations. Such $Z_{HS^2 \times S^1}(\mathcal{B})$ picks up only a phase under the large gauge transformation $x^i \mapsto x^i + 1$ (so that $|Z|^2$ is gauge-invariant), provided that $I_{\mathcal{B}}(x^i + 1) = I_{\mathcal{B}}(x)$, which holds for the half-index. It is possible to interpret $Z_{HS^2 \times S^1}(\mathcal{B})$ as the overlap:

$$Z_{HS^2 \times S^1}(\mathcal{B}) = \langle \mathcal{B}|H \rangle \,, \tag{145}$$

where $\langle \mathcal{B}|$ is a boundary state and $|H\rangle$ is the state created by the hemisphere. Since $Z_{HS^2 \times S^1}(\mathcal{B})$ is a holomorphic section with respect to (143), it is natural to interpret $\langle \mathcal{B}|$ and $|H\rangle$ as holomorphic sections of some line bundles $\mathcal{L}_1$ and $\mathcal{L}_2$, such that $\langle \mathcal{B}|H\rangle$ is a holomorphic section of $\mathcal{L}_1 \otimes \mathcal{L}_2$. In particular, $\langle \mathcal{B}|$ is holomorphic[15] with respect to (143), because the anomaly $\kappa_{ij}$ is carried entirely by the boundary, while $|H\rangle$ must be holomorphic with respect to $\overline{\partial}$. The latter agrees with the fact known from the tt* geometry that the hesmisphere creates states in the holomorphic basis, in which the holomorphic structure is simply $\overline{\nabla} = \overline{\partial}$.

---

[14]The two are related via a more general squashed background [64].

[15]Strictly speaking, just like in Section 3.2, we should be talking about $\langle \mathcal{B}|e^{-\epsilon H}$ rather than $\langle \mathcal{B}|$ here.

- A regularized (via Euclidean evolution) boundary state $e^{-\epsilon H}|\mathcal{B}\rangle$ corresponding to some $\mathcal{N} = (0,2)$ boundary conditions $\mathcal{B}$. Such states generally represent non-trivial $Q$-cohomology classes, same as SUSY vacua. By varying the values of flat flavor connections $x$, the state $e^{-\epsilon H}|\mathcal{B}\rangle$ (or rather its $Q$-cohomology class) assembles into a holomorphic section of a line bundle over $\mathcal{E}_T$. This follows from the same arguments as before: The antiholomorphic parameter $\overline{x}$ couples to some $Q$-exact operator, which fails to decouple only due to the 't Hooft anomaly supported by the boundary. This is most cleanly seen in the case of the interval partition function on $\mathbb{T}^2 \times I$ (here $I = (0, 2\epsilon)$) with the boundary conditions $\mathcal{B}_{1,2}$. On the one hand, it is just an overlap:

$$\langle \mathcal{B}_1 | e^{-2\epsilon H} | \mathcal{B}_2 \rangle. \tag{146}$$

On the other hand, in the IR it is the $\mathbb{T}^2$ partition function of some 2D $\mathcal{N} = (0,2)$ theory. If the boundaries $\mathcal{B}_{1,2}$ support boundary anomalies with the coefficients $k_{ij}^{(1,2)}$, then such a partition function is a holomorphic section with respect to (143), with the total anomaly coefficients $\kappa_{ij} = k_{ij}^{(1)} + k_{ij}^{(2)}$. We can also think of it as a section of $\mathcal{L}_1 \otimes \mathcal{L}_2$, where $\langle \mathcal{B}_1 | e^{-\epsilon H}$ and $e^{-\epsilon H} | \mathcal{B}_2 \rangle$ are holomorphic sections of $\mathcal{L}_1$ and $\mathcal{L}_2$ individually, with the holmorphic structure (143) with $\kappa_{ij} = k_{ij}^{(1)}$ and $\kappa_{ij} = k_{ij}^{(2)}$, respectively.

## 5 Relation to elliptic cohomology

Let $T = U(1)^r$ be the global symmetry preserved by the vacuum, and let $x^i$, as before, parameterize the flat $T$-connections on $\Sigma = \mathbb{T}^2$. If the vacuum remains gapped and isolated for all values of $x \in \mathcal{E}_T$, the above analysis fully applies and we straightforwardly obtain a holomorphic line bundle $\mathcal{L}$ over $\mathcal{E}_T$. This happens, for example, in a class of 3D $\mathcal{N} = 2$ theories whose vacua remain isolated and gapped due to large generic real masses, so that flat connections $x$ do not change the potential and vacua qualitatively. In this case the moduli space of vacua is a point $X = \text{pt}$ (or a discrete set of points), and we identify the space of flat connections with the equivariant elliptic cohomology variety of a point:

$$\text{E}_T(\text{pt}) = \mathcal{E}_T. \tag{147}$$

For the approach to elliptic cohomology that we follow see [66] and references therein. In particular, this $\text{E}_T(\text{pt})$, or more generally $\text{E}_T(X)$, is a variety that generalizes $\text{Spec} H_T(X)$ and $\text{Spec} K_T(X)$ in the cohomological and K-theoretic cases. $\text{E}_T(X)$ is not affine, so there is no corresponding cohomology ring whose Spec it would be. Nonetheless, one can talk about elliptic cohomology classes. Just like the cohomology and K-theory classes can be interpreted as functions on $\text{Spec} H_T(X)$ and $\text{Spec} K_T(X)$, respectively, the elliptic cohomology classes are understood as sections of line bundles on $\text{E}_T(X)$. That is, an elliptic class is given by a line bundle $\mathcal{L}$ on $\text{E}_T(X)$ and its holomorphic section.

More generally, $X$ can be a positive-dimensional moduli space of vacua, such that for generic $x \in \mathcal{E}_T \setminus D$ only a discrete set $X^T = \bigsqcup_a \text{p}_a$ of fixed points stays at zero energy. For example, this happens in the above mentioned class of 3D $\mathcal{N} = 2$ theories, which can be fully gapped via the real masses. Turning off the real masses, we get back the moduli space $X$, which can be again gapped out via the generic flat connections $x$. Physically, $x$ are similar to masses, – they are just doubly-periodic. For non-generic $x \in D$ (discriminant locus) a non-trivial subspace of $X$ remains in the space of vacua (the whole $X$ at $x = 0 \in D$). Suppose an isolated vacuum $\text{p}_a$ stays gapped for $x \in \mathcal{E}_T \setminus D_a$, so

$$D = \bigcup_a D_a. \tag{148}$$

To be more concrete, consider the tangent space $T_{\mathrm{p}_a}X$ at the $a$'th fixed point. It is acted by $T$, so let $\Phi_a$ be the set of weights for this action. An infinitesimal deformation $\delta \in T_{\mathrm{p}_a}X$ of weight $w \in \Phi_a$ away from $\mathrm{p}_a$ acquires a periodic mass $w \cdot x \mod \mathbb{Z} + \tau\mathbb{Z}$. It being zero,

$$w \cdot x = 0 \mod \mathbb{Z} + \tau\mathbb{Z}, \tag{149}$$

is precisely the condition for $\delta$ to represent a flat direction in the space of vacua. Thus we identify the subset of $x$'s for which $\mathrm{p}_a$ fails to be isolated:

$$D_a = \{x \in \mathcal{E}_T \mid \exists w \in \Phi_a, w \cdot x \in \mathbb{Z} + \tau\mathbb{Z}\}. \tag{150}$$

Let $\Delta_{ab} \subset D_a \cap D_b$ denote the locus at which the vacua $a$ and $b$ get reconnected by some $C_{ab} \subset X$. This $\Delta_{ab} \subset \mathcal{E}_T$ parameterizes flat connections for a certain subtorus that will be denoted as $T_{ab} \subset T$.

Each gapped vacuum $\mathrm{p}_a$ has an associated matrix $k_{ij}^{(a)}$ of the effective CS levels for coupling to the background $T$ gauge field. Even though physically the vacuum line bundle (for the vacuum $\mathrm{p}_a$) is only well defined over $\mathcal{E}_T \setminus D_a$, it is clear that it can be extended over the discriminant locus to the entire $\mathcal{E}_T$. Indeed, using the construction discussed earlier, which is based only on the matrix $k_{ij}^{(a)}$, we formally obtain a holomorphic line bundle $\mathcal{L}_a$ with the Berry connection over the entire $r$-dimensional complex torus $\mathcal{E}_T$.

Given two such bundles $\mathcal{L}_a$ and $\mathcal{L}_b$ associated to the vacua $a$ and $b$, let us restrict them to $\Delta_{ab} \subset \mathcal{E}_T$ and prove that they become isomorphic there. If the vacua cannot be reconnected, $\Delta_{ab}$ is empty and the statement is trivially true. If $\Delta_{ab}$ is zero-dimensional (a collection of points), the statement is still obvious because any two line bundles restricted to a point become isomorphic. Now if $\Delta_{ab}$ is a positive-dimensional subtorus, we should note that the data of $\mathcal{L}_a\big|_{\Delta_{ab}}$ is encoded in the Berry connection for flat gauge fields parameterized by $\Delta_{ab}$, which are valued in the subtorus $T_{ab} \subset T$. This Berry connection is computed from the CS levels, as we explained earlier. Now the key point is that the CS terms for vacua $a$ and $b$ become equivalent upon restriction to the subtorus $T_{ab}$, which can be written as follows:

$$\mathrm{CS}^{(a)}\big|_{T_{ab}} = \mathrm{CS}^{(b)}\big|_{T_{ab}}. \tag{151}$$

Restriction to $T_{ab}$ means that we consider the $\mathrm{Lie}(T_{ab})$-valued gauge fields. To understand the equality (151), note that not only each isolated vacuum is equipped with the effective CS levels, but also each connected component of the moduli space of vacua is decorated by the effective CS levels for global symmetries preserved everywhere along that component. Since CS levels are integers, by continuity they remain constant along each connected component. Now because for flat connections $x \in \Delta_{ab}$, i.e., those valued in the subtorus $T_{ab}$, the vacua $a$ and $b$ become reconnected by $C_{ab}$, this means that the effective CS levels $\mathrm{CS}^{(a)}$ and $\mathrm{CS}^{(b)}$, though different on the full torus $T$, must become equal upon restriction to $T_{ab}$, which is precisely the statement (151). Since the CS levels fix the line bundle data, we conclude that $\mathcal{L}_a\big|_{\Delta_{ab}}$ and $\mathcal{L}_b\big|_{\Delta_{ab}}$ are indeed isomorphic.

These statements motivate the following definitions. To each isolated vacuum $\mathrm{p}_a$, associate a copy of $\mathcal{E}_T$, and identify them along the loci $\Delta_{ab}$, resulting in a certain variety. This variety is identical to the (reduction of) the elliptic cohomology scheme:

$$\mathrm{E}_T(X) = \mathcal{E}_T \sqcup \mathcal{E}_T \sqcup \cdots \sqcup \mathcal{E}_T / \sim, \tag{152}$$

where $\sim$ means that the $a$'th and $b$'th copies of $\mathcal{E}_T$ are glued along $\Delta_{ab}$. Since they also carry the line bundles $\mathcal{L}_a$ and $\mathcal{L}_b$, which are isomorphic along $\Delta_{ab}$, we can glue such line bundles into a single $\mathcal{L}$ over $\mathrm{E}_T(X)$. With these data, it makes sense to talk about elliptic cohomology classes, that is holomorphic sections of $\mathcal{L}$. Precisely such a setting plays central role in the constructions of elliptic stable envelopes in mathematics [66] and physics [28, 29]. See also a somewhat different in approach but similar in goals series of papers [67–71].

## 5.1 Chern-Simons couplings in $\mathcal{N} = 4$ theories

For completeness and with the view towards applications [28], let us study the structure of effective CS couplings on the Higgs branch $X$ of 3D $\mathcal{N} = 4$ gauge theories. We look at the effective CS terms for the global symmetry torus $T$, which only make sense in the vacua labeled by the fixed points $X^T = \sqcup_a \mathrm{p}_a$, since other vacua break $T$. Such terms were denoted $\mathrm{CS}^{(a)}$ before. The CS terms for a subtorus $T_{ab} \subset T$ make sense on $X^{T_{ab}}$, which can be larger than $X^T$, and may include positive-dimensional components $C_{ab}$ connecting $\mathrm{p}_a$ and $\mathrm{p}_b$.

### 5.1.1 No additional CS terms from Higgsing

Let us first consider the case without real masses, so the full Higgs branch $X$ is present. Focus on the Higgs vacuum corresponding to an isolated $T$-fixed point $\mathrm{p}_a \in X$, which thus preserves the full torus $T$. Then the dynamical gauge field is locked to a specific value determined by the background gauge fields for global symmetries. For 3D $\mathcal{N} = 4$ gauge theories studied in [28], the global symmetries are $T = U(1)_\hbar \times \mathbf{A} \times \mathbf{A}'$. Here $\mathbf{A}$ is the maximal torus of flavor symmetries acting on the hypermultiplets, $U(1)_\hbar$ is the anti-diagonal subtorus of the maximal torus $U(1)_H \times U(1)_C$ of the R-symmetry $SU(2)_H \times SU(2)_C$, and $\mathbf{A}'$ is the torus of topological symmetries, under which the fields are neutral and the charged objects are monopole operators.[16] The currents for $\mathbf{A}'$ are given by $\star F$ for abelian factors in the gauge group, thus the background gauge field $B_\mu$ for $\mathbf{A}'$ couples to the dynamical fields via a supersymmetric BF term $\mathrm{tr} \int B \wedge F + \ldots$ (see [28, 29, 65] for details, or [72, 73] specifically on the BF terms). At the vacuum $\mathrm{p}_a$, the gauge fields are locked to linear combinations:

$$F^i_{\mu\nu} = \sum_f c^i_f F^f_{\mu\nu} + h^i F^\hbar_{\mu\nu}, \tag{153}$$

where $F^f_{\mu\nu}$ and $F^\hbar_{\mu\nu}$ are gauge field strengths for the $\mathbf{A} \times U(1)_\hbar$ symmetries. The combinations (153) are chosen such that the hypermultiplets that develop vevs in the vacuum $\mathrm{p}_a$ are neutral under the combined $G \times \mathbf{A} \times U(1)_\hbar$ transformations (they are not acted on by $\mathbf{A}'$ anyways). As a result of (153), the BF couplings between $\mathbf{A}'$ and the center of gauge group induce the $\mathbf{A}' \times \mathbf{A}$ and the $\mathbf{A}' \times U(1)_\hbar$ BF couplings (mixed CS terms). Such terms are simply associated with the gauge fields developing vevs, not with integrating out anything, thus they may be called the "classical" part of the effective CS coupling $K^{(a)}$:

$$
\begin{aligned}
K^{(a)} &= K^{(a)}_{\mathrm{cl}} + K^{(a)}_{\mathrm{q}}, \\
K^{(a)}_{\mathrm{cl}} &= \kappa_a + \kappa^C_a,
\end{aligned}
\tag{154}
$$

where we followed the notation from [29] for the $\mathbf{A} \times \mathbf{A}'$ and $U(1)_\hbar \times \mathbf{A}'$ mixed CS levels $\kappa_a$ and $\kappa^C_a$, respectively.

The "quantum" contribution $K^{(a)}_{\mathrm{q}}$ to the matrix of CS levels comes from integrating out massive fields. Since we do not turn on real masses at the moment, the fields become massive through the Higgs mechanism only. Indeed, we are on the Higgs branch after all. In the vacuum $\mathrm{p}_a$, a subset of hypermultiplets remains massless, – they represent the moduli fields in the low-energy theory, and they are not integrated over. These hypermultiplets are charged under the global torus $T$, and naturally they have zero vev in the $T$-invariant vacuum (any vev would break $T$ and move us into the nearby vacuum). The other hypermultiplets develop vevs and, together with the vector multiplets, become massive. In the region of large FI parameters, these fields have large masses and can be integrated out semiclassically.

---

[16]Because $\mathbf{A}'$ does not act on $X$, $\mathrm{E}_T(X) \cong \mathrm{E}_{U(1)_\hbar \times \mathbf{A}}(X) \times \mathcal{E}_{\mathbf{A}'}$, where $\mathrm{E}_{U(1)_\hbar \times \mathbf{A}}(X)$ is often denoted as $\mathrm{Ell}_{U(1)_\hbar \times \mathbf{A}}(X)$. Note that in this paper, $T = U(1)_\hbar \times \mathbf{A} \times \mathbf{A}'$, while in [28] it was $T = U(1)_\hbar \times \mathbf{A}$.

We represent each hypermultiplet as a pair of chirals, and denote the hypers that develop vevs by $(I^i, J_i)$, where the constituent chiral multiplets $I^i$ develop vevs and $J_i$ do not (note that $I^i$ and $J_i$ cannot have vevs simultaneously, so such a splitting is always possible). The CS terms can only be generated at the one loop order,[17] namely via fermionic determinants, so let us focus on the relevant terms involving fermions. The gaugini $\lambda$ from 3D $\mathcal{N}=2$ vectormultiplet and the fermions $\psi_I$ from $I$ together obtain masses via the SUSY kinetic term of the $I$ multiplet. The relevant terms are:

$$-i\overline{\lambda}\slashed{D}\lambda - i\overline{\psi}_I\slashed{D}\psi_I + i\overline{\psi}_I\lambda I - i\overline{I}\overline{\lambda}\psi_I \,. \tag{155}$$

As for the adjoint chiral $\Phi$ (from the 3D $\mathcal{N}=4$ vectormultiplet) and the chirals $J_i$, they obtain mass through the superpotential $W = J\Phi I$. Denoting their fermionic components by $\lambda_\Phi$ and $\Psi_J$, the relevant terms are:

$$-i\overline{\lambda}_\Phi\slashed{D}\lambda_\Phi - i\overline{\psi}_J\slashed{D}\psi_J + i\psi_J\overline{\lambda}_\Phi I - i\overline{\psi}_J\lambda_\Phi\overline{I} \,. \tag{156}$$

Assuming the gauge group is fully Higgsed, all $\dim G$ components of $\lambda$ become massive by pairing with precisely $\dim G$ components of $\psi_I$ in (155), and similarly all $\dim G$ components of $\lambda_\Phi$ receive masses by pairing with the same number of components of $\psi_J$. In this way, the massless vector multiplets double their fermion content, as appropriate for the massive vectors that they become. Explicitly, if $(T^a)_\alpha{}^\beta$ with $a = 1\ldots\dim G$ are the generators of $G$ acting on the (possibly reducible) representation $\mathcal{R} \ni I$, we may write:

$$i\overline{\psi}_I\lambda I = i(\overline{\psi}_I)^\alpha \left[ (T^a)_\alpha{}^\beta I_\beta \right] \lambda^a \,, \tag{157}$$

where we abuse notations and write $I$ for the vev of $I$. Here

$$(\overline{\psi}_I)^\alpha \left[ (T^a)_\alpha{}^\beta I_\beta \right] =: \overline{\Psi}_I^a \,, \tag{158}$$

are precisely the $\dim G$ components of $\overline{\psi}_I$ that become massive. We thus see that (155) really contains a pair of adjoint-valued Dirac fermions, $(\lambda, \overline{\lambda})$ and $(\Psi_I, \overline{\Psi}_I)$, and an off-diagonal mass term $i\overline{\Psi}_I\lambda - i\overline{\lambda}\Psi_I$ for them. This mass matrix can be diagonalized by rotating $\lambda$ with $\Psi_I$, and it has real eigenvalues that come precisely in pairs $\pm m$ for some $m$. Integrating out the (adjoint-valued) fermions associated to the corresponding eigenvectors looks just like integrating out two fermions of opposite real masses, and they of course produce opposite CS terms for $G$ that cancel each other.

The same argument applies to the fermions $(\lambda_\Phi, \overline{\lambda}_\Phi)$ and $(\psi_J, \overline{\psi}_J)$ in (156). The only difference is what gauge fields they couple to. In (155) the fermions $\lambda$ and $\Psi_I$ are effectively in the adjoint representation of G, while in (156) the fermions $\lambda_\Phi$ and the massive components $(\psi_J)^\alpha \left[ (T^a)_\alpha{}^\beta I_\beta \right] = \Psi_J^a$, in addition to being in the adjoint of $G$, also have charge $+1$ under the $U(1)_\hbar$. What matters, however, is that $\lambda_\Phi$ and $\Psi_J$ have the same charges under $G \times U(1)_\hbar$, thus we can still rotate them and diagonalize the mass matrix. Again, the resulting real masses come in pairs, and the CS terms that they generate simply cancel.

We thus see that integrating out the multiplets that become massive purely via the Higgs mechanism generates no CS terms. Hence the effective CS couplings in the fixed point vacuum $p_a$ on the unlifted Higgs branch are equal to their classical values introduced earlier:

$$K^{(a)} = K_{cl}^{(a)} = \kappa_a + \kappa_a^C \,. \tag{159}$$

Once we start turning on flavor connections $x \in \mathcal{E}_T$ on $\mathbb{T}^2$, the potential appears, the Higgs branch gets lifted (except for the fixed points $p_a$), and the vacuum $p_a$ becomes isolated. Even

---

[17]The easiest way to see it is by introducing the loop-counting parameter $\alpha$, known as the Plank constant. Since CS terms are quantized, they cannot be multiplied by continuous parameters, thus they appear at the $\alpha^0$ order, which is precisely the one-loop order.

though the fluctuations away from $p_a$ are massive (with the elliptic masses $w \cdot x$), they do *not* generate any further CS terms. Essentially, this is the result we saw in Section 3, where the Berry connection (given by the magnetic potential of the monopole wall) was trivial for zero real mass and generic $x \in \mathcal{E}_T$. Likewise, here we assume the regularization scheme, in which no CS terms are generated via fermions on $\mathbb{T}^2 \times \mathbb{R}$ that have no real masses but are coupled to flat connections $x$ on $\mathbb{T}^2$ (even though $x$ is very reminiscent of mass).

Thus the line bundles $\mathcal{L}_a$ associated to fixed points on the unlifted Higgs branch are determiend by the classical CS levels (159). Such levels obey (151), confirming that these bundles are pulled back from a single line bundle $\mathcal{L}$ on $\mathrm{E}_T(X)$. This slightly differs from the choice of $\mathcal{L}_a$ in [29] (their CS levels differ by an overall shift).

### 5.1.2 Real masses

Now let us make the fixed point vacua $p_a$ gapped by large real masses, not just the "elliptic masses" $x$. Unlike the latter, the presence of real masses modifies the effective CS levels, because chiral multiplets contain fermions, and fermions of real mass $m$ in the representation $\mathcal{R}$ of some simple $G$ are well known [34–36] to contribute

$$\frac{\mathrm{sgn}(m)}{2} T_{\mathcal{R}} \mathrm{CS}, \tag{160}$$

to the effective action, where CS is the level-one CS functional of the $G$ gauge fields, and $T_{\mathcal{R}}$ is the Dynkin index of $\mathcal{R}$. In principle, scheme-dependent background CS terms could shift CS level in (160) by an $m$-independent integer, without affecting the jump of level at $m = 0$. This jump is scheme-independent and related to the fact that the mass profile $m(y) = m\,\mathrm{sgn}(y)$ in 3d supports a chiral Fermi mode at $y = 0$. The ambiguity of background CS counterterms is easily removed by assuming that the theory possesses boundary conditions.[18] Thinking of a boundary as an interface between our theory and the empty theory, the background CS counterterms do not feel the interface and live in the entire space. It is natural to require that the CS term is identically zero on the empty side. This fixes the CS counterterm ambiguity, and corresponds to the answer given in (160).

Let us compute quantum corrections to the effective CS levels due to the real masses. The vacuum equations include $(\sigma + m)\phi = 0$, where $\phi$ stands for the chiral multiplet scalars, $m$ means real masses, and $\sigma$ – real scalars in dynamical vector multiplets. At the given isolated vacuum, $\sigma$ has a vev that partially screens $m$, such that $\sigma + m$ acts by zero on those components of $\phi$ that develop vevs. Thus such components do not receive any real masses. They still do, together with the vector multiplets, receive masses via the Higgs mechanism, as in the previous subsection. The analysis from that subsection still applies, and in particular, integrating out such multiplets does not generate any CS terms.

The rest of chiral multiplets, namely those acted on nontrivially by $\sigma + m$, receive real masses. These are precisely the multiplets that parameterize normal directions $N_{X/p_a} \cong T_{p_a}X$ to the given isolated fixed point $p_a \in X$. Recall that the set of weights for the $T$-action on $T_{p_a}X$ is denoted $\Phi_a$. The normal bundle to the fixed locus breaks into the attracting and repelling directions, $N_{X/p_a} \cong N_a^> \oplus N_a^<$. The attracting/repelling nature of a given direction is determined by the sign of the effective real mass of the corresponding chiral multiplet. According to this sign, the weights are broken into two subsets $\Phi_a = \Phi_a^+ \cup \Phi_a^-$:

$$\Phi_a^+ = \{w \in \Phi_a, \ \langle w, \sigma_a + m \rangle > 0\}, \tag{161}$$

where $\sigma_a$ is the vev of $\sigma$ in the $a$'th vacuum. Integrating out such massive fields generates the

---

[18]We thank Zohar Komargodski for a discussion of this point.

quantum correction to the effective CS level as in [29]:

$$K_{\mathrm{q}}^{(a)} = \frac{1}{2} \left( \sum_{w \in \Phi_a^+} w \otimes w - \sum_{w \in \Phi_a^-} w \otimes w \right). \tag{162}$$

Indeed, the chiral multiplet corresponding to $w \in \Phi_a^+$ has positive mass and couples to the $T$ gauge field $A$ through the linear combination $A^{(w)} = \langle w, A \rangle$. Integrating out this chiral produces the $\frac{1/2}{4\pi} \int A^{(w)} \mathrm{d}A^{(w)}$ CS term, while a weight $w \in \Phi_a^-$ would contribute $\frac{-1/2}{4\pi} \int A^{(w)} \mathrm{d}A^{(w)}$. Altogether we find (162), where the CS level is understood as an invariant bilinear form on the Lie algebra.

The answer (162) holds even if the $T$-fixed locus is positive-dimensional. For example, this could happen if we turn on the non-generic real masses. In such cases, the weights $\Phi_a^\pm$ in (162) still correspond to the attracting/repelling directions (i.e., chirals of positive/negative effective real masses). The multiplets that remain massless describe the fixed locus, they are not integrated out and hence do not contribute to $K_{\mathrm{q}}^{(a)}$.

Notice also that $\Phi_a^- = \hbar - \Phi_a^+$, because of which

$$K_{\mathrm{q}}^{(a)} = \frac{1}{2} \sum_{w \in \Phi_a^+} (\hbar \otimes w + w \otimes \hbar - \hbar \otimes \hbar). \tag{163}$$

This $K_{\mathrm{q}}^{(a)}$ generally contains half-integral CS levels. To get rid of this inconvenience (recall Section 3), we may redefine the $\hbar$ charges by 2, thus replacing the weight $\hbar$ by $2\hbar$ above.

### 5.1.3 Mini summary

There are Higgs branch vacua corresponding to the isolated fixed points $\mathrm{p}_a \in X$, which preserve the full torus $T = \mathbf{A} \times \mathbf{A}' \times U(1)_\hbar$ of global symmetries. We turn on real masses for some subtorus $\mathbf{A}_1 \subset \mathbf{A}$ of the $\mathcal{N} = 4$ flavor symmetries, with the fixed locus $X^{\mathbf{A}_1}$ not necessarily zero-dimensional. The effective CS terms in the vacuum $\mathrm{p}_a \in X^{\mathbf{A}_1}$ are given by

$$K^{(a)} = K_{\mathrm{cl}}^{(a)} + K_{\mathrm{q}}^{(a)}, \tag{164}$$

where the tree level contribution $K_{\mathrm{cl}}^{(a)}$ (159), as discussed around (153), contains $\mathbf{A} \times \mathbf{A}'$ and $U(1)_\hbar \times \mathbf{A}'$ CS terms resulting from the classical BF term evaluated at the $a$'th vacuum. The one-loop correction $K_{\mathrm{q}}^{(a)}$ is described in (162) and captures effects of the multiplets with real masses, which parameterize the normal bundle $N_{X/X^{\mathbf{A}_1}}$, and whose $T$-weights $\Phi_a = \Phi_a^+ \cup \Phi_a^-$ split into those of positive/negative real masses in $\Phi_a^+$ and $\Phi_a^-$, respectively.

At one extreme, $\mathbf{A}_1 = \mathbf{A}$, we turn on generic real masses, $X^{\mathbf{A}_1}$ is a set of isolated fixed points, and our $K^{(a)}$ agrees with the one in [29]. At anther, $\mathbf{A}_1 = 1$, all real masses vanish, our answer is purely classical, $K^{(a)} = K_{\mathrm{cl}}^{(a)}$, which differs from [29] by the shift of CS levels.

Our CS terms obey the property (151). Thus the bundles $\mathcal{L}_a$ and $\mathcal{L}_b$ over $\mathcal{E}_T$, associated to the vacua $\mathrm{p}_a, \mathrm{p}_b \in X$, agree over the locus $\Delta_{ab} \subset \mathcal{E}_T$ where $\mathrm{p}_a$ and $\mathrm{p}_b$ reconnect, and are pulled back from the line bundle $\mathcal{L}$ on $\mathrm{E}_T(X^{\mathbf{A}_1})$. The vacuum geometry encoded in $\mathcal{L}$ (with the holomorphic structure dictated by the Berry connection) is our main application of the Berry connection in SQFT. Note that relating sections of such bundles for different subtori $\mathbf{A}_2 \subset \mathbf{A}_1 \subset \mathbf{A}$ is done via the ellictic stable envelopes, as discussed extensively in [28, 66].

## 5.2 Higher genus

It is natural to wonder about generalizations of the above, where the higher genus curve $\Sigma$ replaces the elliptic curve of elliptic cohomology, leading to some generalized cohomology.

This was anticipated in Section 4.1, where we replaced $\mathcal{E}_T \equiv \mathcal{E}_T[\mathbb{T}^2]$ by $\mathcal{E}_T[\Sigma] \cong J(\Sigma)^r$ and saw how the effective CS couplings still determine the holomorphic line bundle on $J(\Sigma)^r$.

A single gapped vacuum is the simplest ingredient from this perspective, for it leads to the generalized cohomology of a point. To make it interesting, we need richer spaces to arise as moduli spaces of vacua for the theory on $\Sigma \times \mathbb{R}$. It is known how to achieve this with $\mathcal{N} = 2$ SUSY (or in free theories). More specifically, it requires a holomorphic-topological (HT) twist, such that $\Sigma$ is holomorphic [74–77]. Hilbert spaces $\mathcal{H}[\Sigma]$ of twisted 3D $\mathcal{N} = 2$ theories were studied in [78, 79], and it would be interesting to clarify their relation to the generalized cohomology of moduli spaces. Such a generalized cohomology is not complex-orientable, i.e., lacks the underlying formal group law. However, there are still reasons to believe it is interesting enough, see for example a discussion in [67, Section 6].

# 6 Conclusions

This paper is an amalgamation of various facts, ideas, and applications of the Berry connection in higher-dimensional QFT. The old-fashioned Berry connection [1, 2], along with its recent generalizations [6–9], are all well defined in QFT, as long as the IR issues are treated by making the space compact, and the UV ambiguities are absent. The latter is easy to address by analyzing finite counterterms, as explained in Section 2. We also spent a considerable amount of time on exploring free 3D theories on the torus $\mathbb{T}^2$ in Section 3, and devoted the rest of paper to more general trivially gapped 3D QFTs, as well as relation to the elliptic cohomology as the main application of our techniques.

One interesting direction of the future work is to follow up on the results of Section 2. We found a few QFT setups there, in which the old Berry connection is well defined. It is interesting to explore them further, especially cases of the unambiguous "gravitational Berry connection", i.e., when we vary geometric moduli of the spatial slice $\Sigma$ with time.

Another interesting future direction was mentioned in Section 5.2. It involves clarifying the relation between twisted Hilbert spaces of 3D $\mathcal{N} = 2$ theories on the higher genus curves $\Sigma$, the corresponding spaces of SUSY vacua, and the generalized cohomolory theories.

# Acknowledgments

This work originated as a spin-off of discussions with N. Nekrasov on the larger program that includes [28] and other works to follow. The author also benefited a lot from discussions and/or correspondence with: A. Abanov, T. Dimofte, A. Ferrari, J. Hilburn, A. Kapustin, Z. Komargodski, M. Litvinov, N. Sopenko.

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
