# Peer review of "Remarks on Berry Connection in QFT, Anomalies, and Applications"

_SciPost Physics, doi:SciPost Phys. 15, 167 (2023)_

## Round 2 · Referee Report · Lev Spodyneiko · 2023-7-10

Report

The paper discussed the definition and applications of Berry connection in the framework of QFT. The author thoroughly studies complications and ambiguities related to it. The discussion smoothly evolves from a general QFT to more specific examples of supersymmetric theories. The paper provides a novel and synergetic link between different research areas and thus satisfies the publication criteria. I have some comments/suggestions/questions:

1. The IR divergences are claimed to be fixed by considering the theory on a compact space $\Sigma$. However, the effective action (say of the example in Sec. 3) does not scale with the volume of $\Sigma$. This is counter-intuitive because one naively would expect that it should scale with the number of degrees of freedom.

2. A recent paper (https://arxiv.org/abs/2305.06399v1) discussed an interpretation of a higher Thouless pump in 2+1D. They have shown that it gives the Berry connection of a mode localized on fluxon insertion. Namely, the Berry curvature of the 2+1 D system is infinite as well as the Berry curvature of the same system with fluxon insertion. However, their difference is finite and well-defined. The discussion of Sec 2 and 3 of the present paper seem to be similar because Berry curvature couples only to the holonomy of the fields on $\Sigma$. I think it is a good idea to discuss the relationship. (The mentioned paper was published after the present one, so I can't insist on it.)

3. Points 1 and 2 above, make me think that the author actually discusses higher Berry curvatures. Namely, I would call "old-fashioned" Berry curvature the one which scales with the volume. If it does not, I would expect that it is actually the Berry curvature of a single mode, or more precisely the difference of Berry curvature of the whole system with and without the mode.

4. I think it is a good idea to explicitly discuss how different issues raised in Sec 2 affect the simple example of Sec 3. For example, how the counterterm of eq. (2.6) appears in the analysis of Sec 3 when $F\ne0$? As far as I understand the other counterterm (2.9) is discussed in the last paragraph of Sec 3.1. If it is, then the author should clarify this by explicit reference to this equation.

And some minor corrections:

5. There is an empty reference [] in the first paragraph of Sec 2.4.

6. $B^{(p)}_i$ in eq. (3.48) is not defined.

Requested changes

Please address 1,4,5,6 from above, and I would appreciate the author's thoughts on 2,3.

---

## Round 2 · Referee Report · Anonymous · 2023-8-2

Report

This work studied the question whether the conventional Berry connection in quantum mechanics is well-defined when we compactify a higher-dimensional QFT on a compact space down to a quantum mechanics. It addressed the question by classifying possible counterterms that can potentially spoil the well-defineness of the Berry connection. Then it discussed the applications of Berry connections in various examples.

Overall, it is a solid work that clarifies the use of Berry connections in higher-dimensional QFTs and points out interesting relations between Berry connections and various old results in the literature. The referee recommends the publication of this manuscript.

Requested changes

One minor comment:

1. A citation in the first paragraph on page 18 is missing.

---

## Round 3 · Referee Report · Lev Spodyneiko · 2023-9-6

Report

The author addressed my questions in the revised version. I recommend publication.

---

## Round 3 · List of Changes

Questions 1,4,5,6 from the Report 1 (which also includes a comment from the Report 2) have been addressed.

1) New footnote 4 is meant to address the first question. This is also related to question 3. Indeed, in Section 3 the Berry connection is not proportional to volume, simply because it can be understood as descending from the topological term (CS term) in the 3d effective action. In other words, indeed, the example of Section 3 may be also seen as the higher Berry connection. In this paper, however, we study the "old-fashioned" Berry connection viewpoint, in the sense that we only allow parameters to vary in time, while remaining constant along the (compact) space. This mixes the"old" and "new" phenomena in the resulting 1d effective action, and a priori one does not assume any volume scaling.

2) As mentioned by the first referee, answering his question 2 is optional, since it refers to the paper 2305.06399 that appeared one year later than the current manuscript. We believe the burned is on the authors of 2305.06399 to make this comparison. Superficially, however, the connection is not immediate, since in Section 3 we look at the Chern-Simons term, which is different from the Thouless pump.

3) To address question 4, the first paragraph of Section 3 has been modified (which also includes a new footnote 9). Also, slightly modified the last paragraph of Subsection 3.1, as requested.

4) Missing reference added (as per comment 5 of the first referee and the comment of the second referee).

5) Paragraph after eqn. (3.48) modified to address the question 6.

You are currently on this page

Resubmission 2211.15680v3 on 6 September 2023

---

## Editorial Decision

published